EMBO
Molecular Medicine

# Liver protects neuron viability and electrocortical activity in post-cardiac arrest brain injury

Zhiyong Guo [1,2,3,11✉], Meixian Yin [1,2,3,4,11], Chengjun Sun[1,2,3,5,11], Guixing Xu[6,11], Tielong Wang[1,2,3,11], Zehua Jia[1,2,3], Zhiheng Zhang[1,2,3], Caihui Zhu[1,2,3], Donghua Zheng[7], Linhe Wang[1,2,3], Shanzhou Huang[1,2,3], Di Liu [1,2,3], Yixi Zhang[1,2,3], Rongxing Xie[1,2,3], Ningxin Gao[1,2,3], Liqiang Zhan[1,2,3], Shujiao He[1,2,3], Yifan Zhu [1,2,3], Yuexin Li[1,2,3], Björn Nashan [8], Schlegel Andrea[9], Jin Xu[10], Qiang Zhao [1,2✉] & Xiaoshun He [1,2✉]

## Abstract

Brain injury is the leading cause of mortality among patients who survive cardiac arrest (CA). Clinical studies have shown that the presence of post-CA hypoxic hepatitis or pre-CA liver disease is associated with increased mortality and inferior neurological recovery. In our in vivo global cerebral ischemia model, we observed a larger infarct area, elevated tissue injury scores, and increased intravascular CD45+ cell adhesion in reperfused brains with simultaneous hepatic ischemia than in those without it. In the ex vivo brain normothermic machine perfusion (NMP) model, we demonstrated that addition of a functioning liver to the brain NMP circuit significantly reduced post-CA brain injury, increased neuronal viability, and improved electrocortical activity. Furthermore, significant alterations were observed in both the transcriptome and metabolome in the presence or absence of hepatic ischemia. Our study highlights the crucial role of the liver in the pathogenesis of post-CA brain injury.

**Keywords** Brain Injury; Cardiac Arrest; Normothermic Machine Perfusion; Liver Dysfunction; Ketone Body Production
**Subject Categories** Cardiovascular System; Neuroscience; Vascular Biology & Angiogenesis

## Introduction

Sudden cardiac arrest (CA) refers to an unexpected cessation of heart function that can lead to death and remains a major public health issue, accounting for ~50% of all cardiovascular-related fatalities (Al-Khatib, 2018). Each year, an estimated 375,000–700,000 citizens experience CA in Europe and the United States (Berdowski et al, 2010; Mozaffarian et al, 2016). Despite advancements in cardio-pulmonary resuscitation (CPR) and life support techniques, post-CA survival rates remain poor, varying between 8 and 23% (Atwood et al, 2005; Girotra et al, 2012; Meaney et al, 2010; Nichol et al, 2008).

Mortality after CA is mainly triggered by post-CA shock and brain injury (Laver et al, 2004). Following the return of spontaneous circulation, the whole-body ischemia-reperfusion response leads to the post-CA syndrome, comprising post-CA brain injury, myocardial dysfunction, systemic ischemia-reperfusion response, and persistent precipitating pathology, resulting in significant morbidity and mortality (Neumar et al, 2008). Indeed, post-CA multiorgan dysfunction largely affects the recovery of brain injury (Neumar et al, 2008). Nevertheless, the specific roles of individual organs other than the heart and circulation in post-CA brain injury are largely unknown, limiting the development of novel therapeutic strategies.

Studies have shown that hypoxic hepatitis (HH) occurs in 7–21% of post-CA patients and is significantly associated with poor neurological outcomes, high mortality rates and prolonged stay in the intensive care unit (ICU) (Champigneulle et al, 2016; Iesu et al, 2018; Roedl et al, 2019; Roedl et al, 2017). On the other hand, it has been demonstrated that CA survivors with comorbid cirrhosis experience worse outcomes compared to those without. Particularly, no patients with Child-Turcotte-Pugh C cirrhosis and advanced acute-on-chronic liver failure survived longer than 28 days with a favorable neurological outcome (Roedl et al, 2017). Similarly, the results from a retrospective observation study, using a nationwide population-based out-of-hospital cardiac arrest (OHCA) registry, have also shown that the overall clinical and neurological outcomes are poorer in OHCA patients with liver cirrhosis than in those without it (Pak et al, 2021). We therefore hypothesize that the liver might play a crucial role in the pathogenesis of post-CA brain injury.

In the current study, results from the in vivo global cerebral ischemia model show that ischemic brain injuries are exacerbated

[1]Guangdong Provincial International Cooperation Base of Science and Technology, Guangzhou, China. [2]Guangdong Provincial Key Laboratory of Organ Medicine, Guangzhou, China. [3]NHC Key Laboratory of Assisted Circulation (Sun Yat-sen University), Guangzhou, China. [4]Department of Anatomy and Neurobiology, Zhongshan School of Medicine, Sun Yat-sen University, Guangzhou, China. [5]Department of Organ Transplantation, Guangdong Provincial People's Hospital (Guangdong Academy of Medical Sciences), Southern Medical University, Guangzhou, China. [6]Department of Neurosurgery, The First Affiliated Hospital, Sun Yat-sen University, Guangzhou, China. [7]Intensive Care Unit, The First Affiliated Hospital, Sun Yat-sen University, Guangzhou, China. [8]Organ Transplant Center, The First Affiliated Hospital of the University of Science and Technology of China, Hefei, China. [9]General and Liver Transplant Surgery Unit, Fondazione IRCCS Ca' Granda Ospedale Maggiore Policlinico, Milan, Italy. [10]State Key Laboratory of Biocontrol, School of Life Sciences, Sun Yat-Sen University, Guangzhou, China. [11]These authors contributed equally as first authors: Zhiyong Guo, Meixian Yin, Chengjun Sun, Guixing Xu, Tielong Wang. ✉E-mail: guozhiy2@mail.sysu.edu.cn; zhaoq37@mail.sysu.edu.cn; hexsh@mail.sysu.edu.cn

in the presence of simultaneous liver ischemia are exacerbated compared to those without liver ischemia. In addition, the results of the ex vivo brain normothermic machine perfusion (NMP) model demonstrate that a well-functioning liver can protect against post-CA brain injury. These results of the current study provide direct evidence supporting the crucial role of the liver in the pathogenesis of post-CA brain injury.

# Results

## Concurrent hepatic ischemia aggravates brain injuries in the global cerebral ischemia model

To investigate how the liver might affect post-CA brain injury, we established an in vivo 30-min global cerebral ischemia model simultaneous hepatic ischemia (BLWI-30) and without hepatic ischemia (BWI-30) (Fig. 1A,B). Cerebral tissue oxygen saturation ($SctO_2$) remained stable in the Sham group, whereas it declined in the BLWI-30 and BWI-30 groups after the blood supply to the brain was obstructed (Appendix Fig. S1A). At 6 h after reperfusion, the pigs in the BLWI-30 and BWI-30 groups had higher neurological severity scores compared to those in the Sham group (Appendix Fig. S1B). In addition, serum levels of aspartate transaminase (AST) and lactate dehydrogenase (LDH) levels at 4 h after reperfusion were higher (Appendix Fig. S1C,D), and liver tissue damages, characterized by hepatocyte swelling and centri-lobular hepatocyte cell membrane rupture, were more obvious at 24 h after reperfusion (Appendix Fig. S1E,F) in the BLWI-30 group compared to the BWI-30 group. The 24-h mortality rate was 28.57% (2/7) in both the BWI-30 and BLWI-30 groups. No surgical complication was documented. These data indicate successful establishment of the global cerebral ischemia model, both with and without simultaneous hepatic ischemia has been established successfully.

To assess the impact of the liver on ischemia reperfusion injury in the brain, we conducted Triphenyl-tetrazolium chloride (TTC) staining to evaluate the infarct area of the frontal lobe, hematoxylin-eosin (HE) staining to assess tissue injury score in different areas of the brain, Nissl staining to assess the viability of the neurons, and measured the expression of genes *IL6* and *TIMP1* to assess blood-brain barrier (BBB) integrity. TTC staining of the frontal lobe showed that the infarct ratio was higher in the BLWI-30 group than in the BWI-30 group (Fig. 1C,D). Similarly, the tissue injury scores in the frontal lobe, occipital lobe, and CA1 region of the hippocampus were higher in the BLWI-30 group than in the BWI-30 group (Fig. 1E–K). As expected, the average tissue injury score in the brain (average score of the frontal lobe, parietal lobe, temporal lobe, occipital lobe, and hippocampus) was much higher in the BLWI-30 group than in the BWI-30 group (Fig. 1L).

The hippocampus is more susceptible to ischemic injury than other brain regions (Johnson, 2023; Zhang et al, 2022). The Nissl staining revealed a lower number of live neurons in the dentate gyrus of the hippocampus in the BLWI-30 group compared to the BWI-30 group. The number of live neurons in the CA1 region of the hippocampus was also lower in the BLWI-30 group compared to the BWI-30 group, although the difference was not statistically significant (Fig. 1M–O). Furthermore, we detected the expression of genes (*TIMP1*, *IL6*, *MMP9*, and *AQP4*) related to blood-brain barrier (BBB) damage (Appendix

Fig. S2, Appendix Table S1). The expression of *IL6* in the frontal lobe tissue was higher in the BLWI-30 group than in the BWI-30 group (Appendix Fig. S2B). The expression of *TIMP1* and *IL6* (Appendix Fig. S2E,F) in the temporal lobe tissues were higher in the BLWI-30 group than in the BWI-30 group. Taken together, these results indicate that ischemia reperfusion injury of the brain is exacerbated by simultaneous liver ischemia.

## The number of intravascular CD45+ cells increase in reperfused brains with concurrent liver ischemia

Damage to the BBB triggers neuroinflammation (Johnson, 2023). Studies have shown increased immune cell infiltration in the brain tissue during ischemia-reperfusion injury (Dal-Bianco et al, 2016; Gelderblom et al, 2009). To determine whether concurrent liver ischemia can induce an increase in the number of immune cells infiltrating the brain, we performed HE and immunohistochemical staining. In HE-stained sections of the five brain regions, more nucleated blood cells were found in the vascular lumen in the brain, particularly in the frontal lobe of the BLWI-30 group compared to the BWI-30 group (Fig. 2A–G). Further characterization of nucleated hematopoietic cells through immunohistochemical staining of CD45 revealed more CD45+ cells in the brain, particularly in the frontal and temporal lobes of the BLWI-30 group compared to the BWI-30 group (Fig. 2H–N). These results suggest that the ischemic liver might increase the local immune response during brain ischemia-reperfusion injury.

## Brain circulation and metabolic activity post-CA is restored during ex vivo brain NMP

In the in vivo global cerebral ischemia model, clamping of the portal vein transiently affected the hemodynamic stability (Fig. EV1A) and serum lactate levels (Fig. EV1B,C), which might affect the extent of brain injury. Recently, an ex vivo brain NMP model has been developed to investigate the pathogenesis of brain diseases (Vrselja et al, 2019). To comprehensively evaluate how the liver impacts the recovery of post-CA brain injury, we constructed ex vivo models: one focusing solely on the brain and the other incorporating liver-assisted brain NMP. In these models, the brain suffered complete, rather than global, ischemia before reperfusion, without affecting the hemodynamic stability by surgical manipulation of the liver. The pressure-controlled NMP provided normothermic, oxygenated perfusion to the brain with whole blood-based perfusate to mimic the return of spontaneous circulation post-CA. The perfusion flow, metabolic activity and electrocortical activity were monitored during ex vivo NMP. Figure 3A, Appendix Fig. S3A, and Appendix Table S2 demonstrate the technology for the liver-assisted brain NMP. There were six experimental groups in the current study: a brain-only control with rapid NMP (BOR group), a liver-assisted brain with rapid NMP (LABR group), and four liver-assisted brain groups in which brain NMP was preceded by 30–240 min of warm ischemia time (WIT) (LABWI groups) (Appendix Fig. S3A). Brains isolated under identical conditions but without undergoing NMP were used as the Sham controls in our study.

Due to the preparation of whole blood-based perfusate, the brains in the BOR group underwent a WIT of $10.1 \pm 1.8$ min (Appendix Fig. S3A). Because of the additional procedures, which

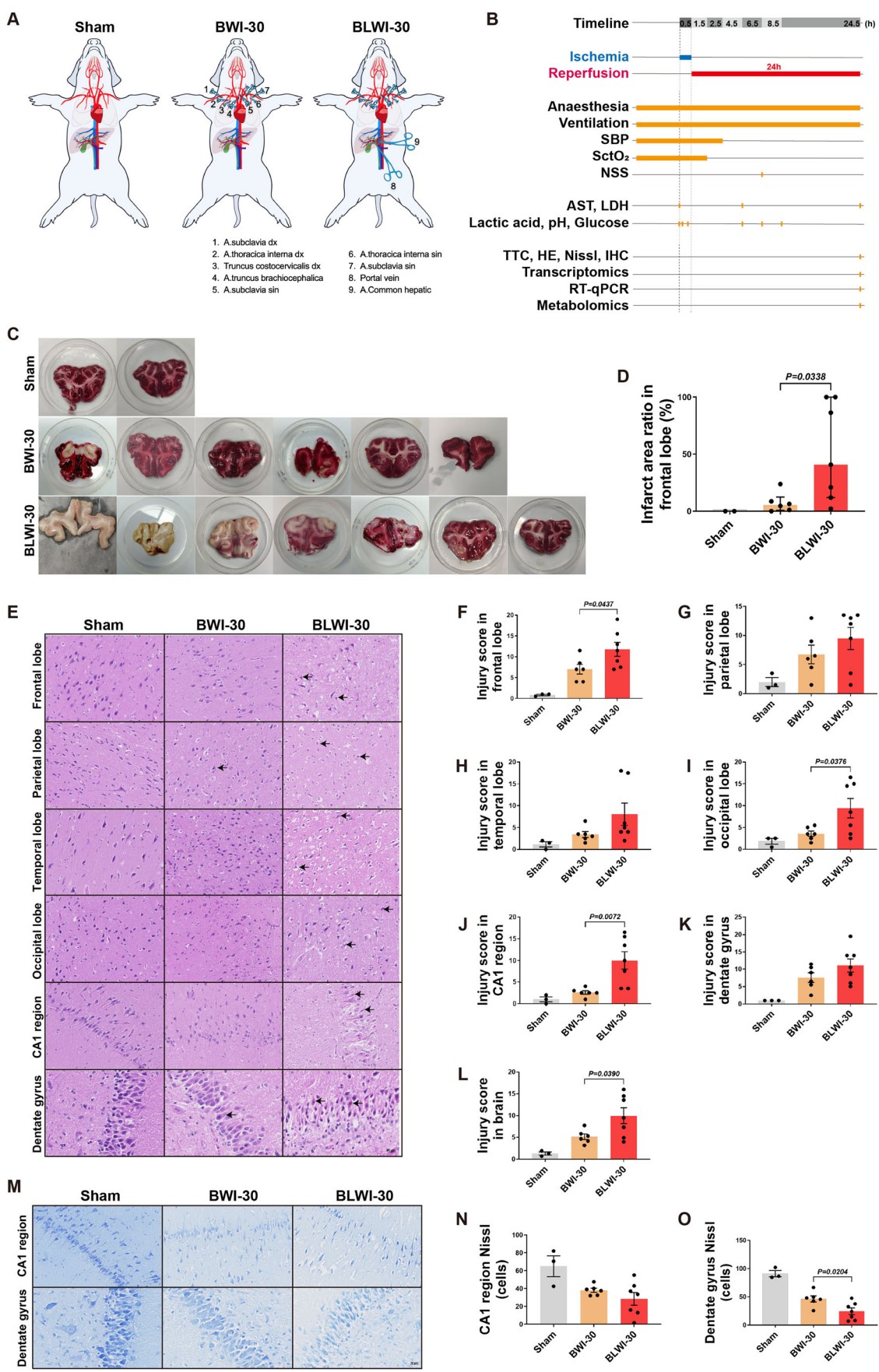

Figure 1. Comparative analysis of brain injuries: greater severity in the BLWI-30 group compared to BWI-30.

**(A)** Schematic of the in vivo pig model of the Sham (no ischemia), BWI-30 (brain with 30-min warm ischemia), and BLWI-30 (brain and liver with 30-min warm ischemia) groups. **(B)** Timeline of the experimental workflow. SBP, systolic blood pressure; SctO$_2$, cerebral tissue oxygen saturation; NSS, neurological severity scores; AST, aspartate transaminase; LDH, lactate dehydrogenase; TTC, triphenyl-tetrazolium chloride; HE, hematoxylin-eosin; IHC, immunohistochemistry; RT-qPCR, quantitative reverse transcription-polymerase chain reaction. **(C)** TTC staining of the frontal lobe. The white area indicates infarct tissue. **(D)** The infarct area ratio in the frontal lobe of three groups. Sham, n = 2; BWI-30, n = 6; BLWI-30, n = 7; unpaired two-tailed ratio Mann–Whitney test; histogram, median with interquartile range. **(E)** Hematoxylin and eosin staining of the frontal lobe, parietal lobe, temporal lobe, occipital lobe, CA1 region of hippocampus (100×), and dentate gyrus of hippocampus (200×). The black arrowhead pointed to the red neuron. **(F–L)** The injury scores were calculated as the mean of two fields in the frontal lobe, parietal lobe, temporal lobe, occipital lobe, CA1 region of the hippocampus, and dentate gyrus of the hippocampus and the brain (average score of the frontal lobe, parietal lobe, temporal lobe, occipital lobe and hippocampus). **(M)** Nissl staining in the CA1 region (100×) and dentate gyrus (200×) of the hippocampus. **(N, O)** Live neuron counts of the two regions. **(F–L)**, **(N, O)** Sham, n = 3; BWI-30, n = 6; BLWI-30, n = 7; all replicates shown were biological replicates; Mean ± SEM, two-tailed ratio unpaired *t*-test. Source data are available online for this figure.

included procuring the liver and connecting it to the system, the mean WIT of the brains was about 4 min longer in the LABR group (14.2 ± 0.7 min) than in the BOR group (P = 0.0670) (Appendix Fig. S3A). The livers continued to produce bile during perfusion with a portal flow exceeding 500 mL/min. They exhibited a homogeneous appearance with parenchyma tissue of soft consistency throughout the procedure, indicating that the livers were functioning well. Ex vivo circulation and oxygen consumption were successfully restored in both the BOR and LABR groups. There was no significant difference in the perfusion pressure, pO$_2$, pCO$_2$, pH, as well as lactate and glucose levels between the BOR and LABR groups during ex vivo NMP (Appendix Fig. S3B–G). However, the left middle cerebral artery (MCA) flow declined in the BOR group, while it maintained stable in the LABR group (Appendix Fig. S3H). Consistently, the arterial resistance declined during the first 2.5 h but increased thereafter in the BOR group. In contrast, the arterial resistance remained stable in the LABR group (Appendix Fig. S3I).

Collectively, we have established both ex vivo brain-only NMP and liver-assisted brain NMP models. These models enable the restoration of circulation and metabolic activity in the brains ex vivo after CA.

## Liver-assisted brain NMP reduces post-CA brain injury

Next, we tested how the inclusion of the liver in the ex vivo NMP circuit could impact the recovery of post-CA brain injury.

In line with the increased arterial resistance and reduced perfusion flow, obvious edema of the reperfused brains was observed after 6-h NMP in the BOR group, while edema was not obvious in the LABR group (Fig. 3B). To assess ischemic brain injuries, HE staining of different brain areas was conducted and tissue injury scores were calculated. In the BOR group, notable pathological changes were observed, including shrunken cells, pyknotic nuclei, and expanded perinuclear space in cortical and hippocampal cells, compared to those in the Sham group. In contrast, in the LABR group, pyramidal neurons exhibited plump cell bodies with large, central nuclei and abundant nerve fibers, similar to those observed in the Sham group (Fig. 3C). The tissue injury scores in the brains, particularly in the cortex and hippocampus, were lower in the LABR group than in the BOR group (Fig. 3D–I).

In addition, the liver-assisted perfused brains had a substantially decreased perfusate level of S100-β, a biomarker of neural injury (Michetti et al, 2019), compared to the brains in the BOR group (Fig. 3J). Moreover, immunohistochemistry (IHC) was

performed on the cerebral cortex to evaluate ischemia-reperfusion injury using markers such as hypoxia inducible factor-1α (HIF-1α), heat shock protein 70 (HSP70), antioxygen nuclear factor erythroid 2 like 2 (NRF2) and DNA repairing 8-oxoguanine DNA glycosylase (OGG). The results demonstrated that the brains in the BOR group had a higher density of these markers compared to both the Sham and LABR groups (Fig. 3K–M; Appendix Fig. S4), suggesting that ischemia-reperfusion injury of the brain is aggravated when the liver is absent from the NMP circuit.

Taken together, these results suggest that the liver can protect the brain from post-CA ischemia-reperfusion injury.

## Liver-assisted brain NMP improves neuronal viability and protects the cytoarchitecture

Considering that the concurrent ischemia of the liver increased the infarct area of the frontal lobe in the in vivo global cerebral ischemia model, we then assessed the density of live neurons in the ex vivo model using Nissl staining. The staining revealed comparable densities of live neurons between the Sham and LABR groups in both the CA1 region and dentate gyrus of the hippocampus. In contrast, the density of live neurons in the CA1 region was lower in the BOR group than in the LABR group (Fig. 4A–C). Immunofluorescence analysis of a microglial marker ionized calcium binding adapter molecule 1 (IBA1) produced fragmented signals, with signs of cellular destruction in the CA1 region of the BOR brains but preserved density in the dentate gyrus of all perfused brains in the LABR group (Appendix Fig. S5A–C). Staining for the astrocytic marker glial fibrillary acidic protein (GFAP) revealed preserved astrocyte density in all three groups (Appendix Fig. S5D–F). Collectively, incorporating the liver into the ex vivo brain NMP setup can protect the reperfused brains from cell death in the neurons and microglia within the CA1 region.

To further evaluate the ultrastructural characteristics of reperfused brains, we conducted the transmission electron microscopy analysis on the brain tissues. In the cerebral cortex and hippocampus of the BOR group, we observed severe tissue vacuolization, mild microvascular edema, slightly loosened myelin sheath, extensive mitochondrial swelling or membrane rupture, rupture and degradation of mitochondrial cristae, and significant synaptic degradation. In contrast, the microvessels, neurites, myelin sheaths, mitochondria, synapses, and neurotransmitters were structurally intact in the majority of cells in the LABR group. Erythrocytes were observed in the microvascular lumen of the LABR and Sham groups, whereas granulocytes were observed in the BOR group (Fig. 4D,E), which is in line with in vivo studies that

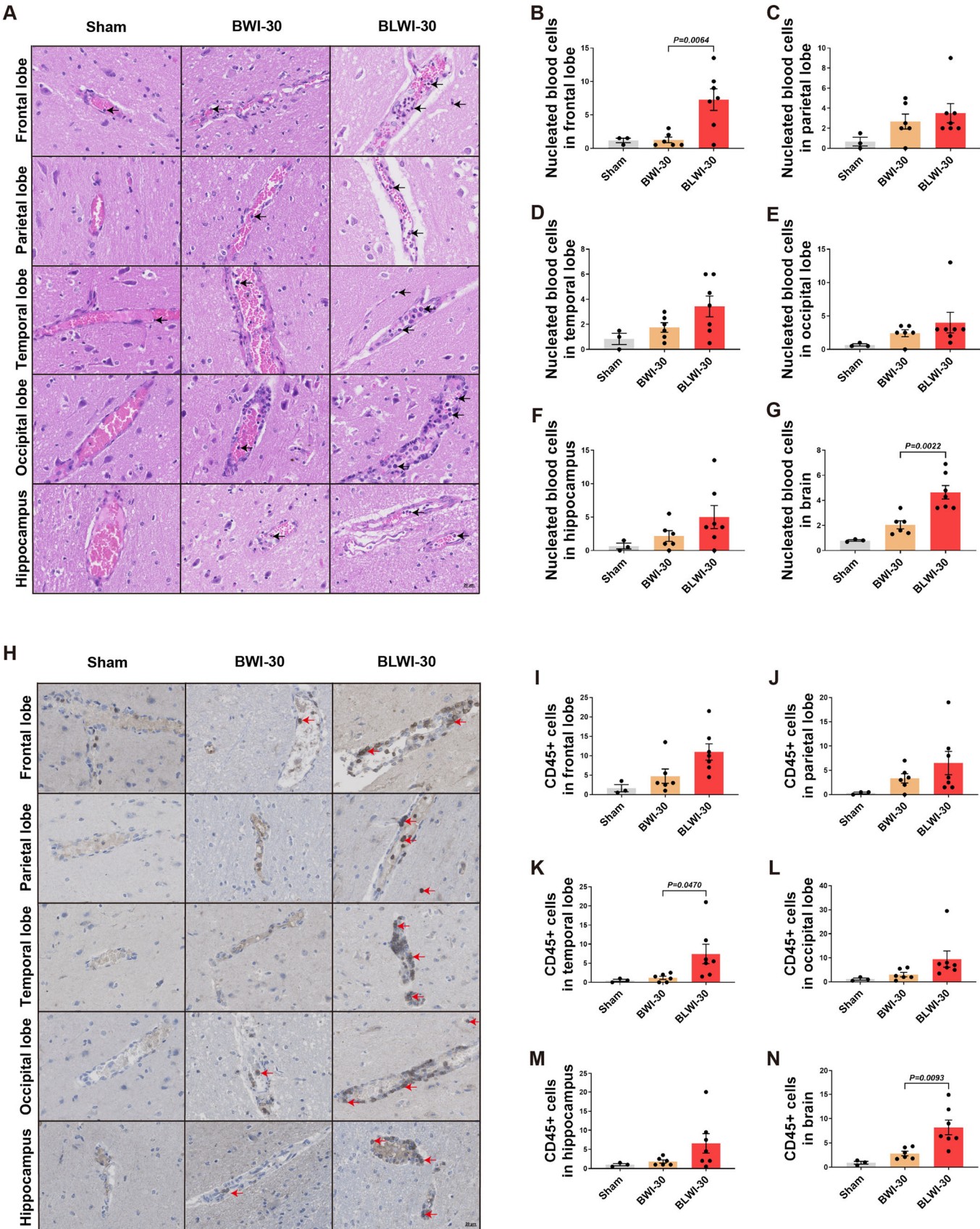

◄

**Figure 2.   A greater number of intravascular CD45-positive cells were observed in the BLWI-30 group compared to the BWI-30 group in vivo.**

**(A)** Hematoxylin-eosin staining showed vascular and nucleated blood cells in the frontal lobe, parietal lobe, temporal lobe, occipital lobe, and hippocampus of pigs (200×). The arrowhead pointed to nucleated blood cells. **(B–G)** The number of nucleated blood cells per field in the brain (average number in the frontal lobe, parietal lobe, temporal lobe, occipital lobe, and hippocampus). **(H)** Immunofluorescence staining for CD45 in the frontal lobe, parietal lobe, temporal lobe, occipital lobe, and the hippocampus of pigs (200×). The arrowhead pointed to brown-yellow CD45+ cells. **(I–N)** The mean number of CD45+ cells of two fields in the frontal lobe, parietal lobe, temporal lobe, occipital lobe, hippocampus or brain (average number of the frontal lobe, parietal lobe, temporal lobe, occipital lobe, and hippocampus). **(B–G)**, and **(I–N)** Sham, no ischemia, $n = 3$; BWI-30, brain with 30-min warm ischemia, $n = 6$; BLWI-30, brain and liver with 30-min warm ischemia, $n = 7$; all replicates shown were biological replicates; Mean ± SEM, two-tailed ratio unpaired *t*-test. Source data are available online for this figure.

observed more CD45+ cells trapped in the vessels of the brains with concurrent liver ischemia.

Collectively, integrating a liver into the ex vivo brain NMP circuit can enhance neuronal viability and maintain cytoarchitecture.

## Liver-assisted brain NMP preserves global electrophysiological activity

It is crucial to determine if the reduction in brain injury translates to improved global electrophysiological activity in the LABR group. Global electroencephalograph (EEG) monitoring revealed the presence of α and/or β waves, both of which are considered to represent conscious activity (Koch et al, 2016; Mashour and Hudetz, 2018), within 30 min after the start of NMP. However, these waves disappeared after 3–5 h of NMP in four of five brains in the BOR group (Fig. 4F; Appendix Table S3). In contrast, frequent α and β waves appeared within 1 h of NMP and persisted throughout the entire 6-h NMP period in all the perfused brains in the LABR group (Fig. 4F; Appendix Table S3). At the end of NMP, the EEG scores and spectroscopic entropy were much higher in the LABR group than in the BOR group (Fig. 4G,H). Collectively, these data show that electrocortical activity can be maintained ex vivo by liver-assisted brain NMP instead of brain-only NMP.

To determine the maximum duration of warm ischemia after which global electrophysiological activity can be restored and maintained ex vivo in pigs, we extended the WIT of the brains to 30 min (LABWI-30), 50 min (LABWI-50), 60 min (LABWI-60), or 240 min (LABWI-240) with liver support (Fig. EV2). Global electrophysiological activity was restored in all five brains, and the presence of α or β waves was maintained until the end of NMP in four of five brains in the LABWI-30 group and in all five brains in the LABWI-50 group (Appendix Table S3). In the LABWI-60 group, all brains exhibited α or β waves during 3–4 h of NMP, although the waves disappeared shortly thereafter (Appendix Table S3). In the LABWI-240 group, no α or β waves were observed (Appendix Table S3). These results indicate that global electrophysiological activity of the brain can be restored and maintained ex vivo following a WIT of over 50 min when a functioning liver is incorporated into the NMP circuit.

## Transcriptome analysis indicates decreased cell death and immune response in brain tissues without simultaneous hepatic ischemia compared to those with the condition

To elucidate the molecular mechanisms underlying the protective effects of the liver on brain injury, RNA-seq was employed to profile the frontal and temporal lobes of the brain with or without concurrent hepatic ischemia. In the frontal lobe, we observed a significant

upregulation and downregulation of genes in the presence of simultaneous hepatic ischemia. We identified 245 upregulated genes and 350 downregulated genes in the frontal lobe with simultaneous hepatic ischemia (BLWI-30 group) (Fig. 5A). Functional annotation of the differentially expressed genes highlighted distinct functional terms. The upregulated genes in the BLWI-30 group were enriched in GO terms related to the regulation of programmed cell death and immune response, such as positive regulation of apoptotic process, regulation of T cell chemotaxis, and B cell activation. Conversely, genes downregulated in the BLWI-30 group were enriched in GO terms associated with neuronal functions, including neuron projection development and regulation of neurotransmitter levels (Fig. 5B). Similar transcriptome differences were observed in the temporal lobe, with 269 upregulated genes and 337 downregulated genes in the BLWI-30 group (Fig. 5C). These differentially expressed genes exhibited comparable enrichment in functional gene ontology terms (Fig. 5D). These molecular alterations corroborate our prior findings on brain function and pathological analyses, indicating that the support of a functioning liver can reduce post-CA brain injury.

Furthermore, we identified differential genes involved in metabolic processes. For instance, genes such as *AK7*, *AK9*, *TYMS*, involved in nucleotide metabolism, may be influenced by cell death and immune responses (Fig. EV3). Intriguingly, we also identified differentially expressed genes enriched in terms related to the response to energy reserve metabolic processes, including genes involved in lipid, linoleic (*PLA2G2D*), and ketone (*BMP5*, *PTGDR2*) metabolism processes (Fig. 5E), as well as in glycolysis/gluconeogenesis (*ALDOB/PCK1*) (Fig. EV3). These results suggest that the liver might regulate the post-CA brain injury through altering metabolic activity of the brain.

## Dramatic metabolome differences in brain tissues with and without simultaneous hepatic ischemia were revealed

To unveil the distinct metabolic dynamics during brain ischemic injury with or without liver ischemia, we conducted an ultra-high-performance liquid chromatography quadrupole time-of-flight mass spectrometry (UHPLC-QTOFMS) analysis of the frontal and temporal lobes. Supervised partial least squares analysis (PLS-DA) of the overall metabolic profiles from 5–7 biological replicates in both groups clearly demonstrated distinct metabolomes (Figs. 6A,B and EV4A,B). The general distribution of metabolism classes is summarized in Fig. EV4C,D.

Interestingly, we found that the downregulated metabolites in the BLWI-30 group, which were higher in the BWI-30 group, were significantly enriched in pathways such as pantothenate and CoA biosynthesis, and glycerophospholipid metabolism in both brain regions (Fig. 6C,D). Further analysis revealed that metabolites enriched in the linoleic acid, arachidonic acid, and

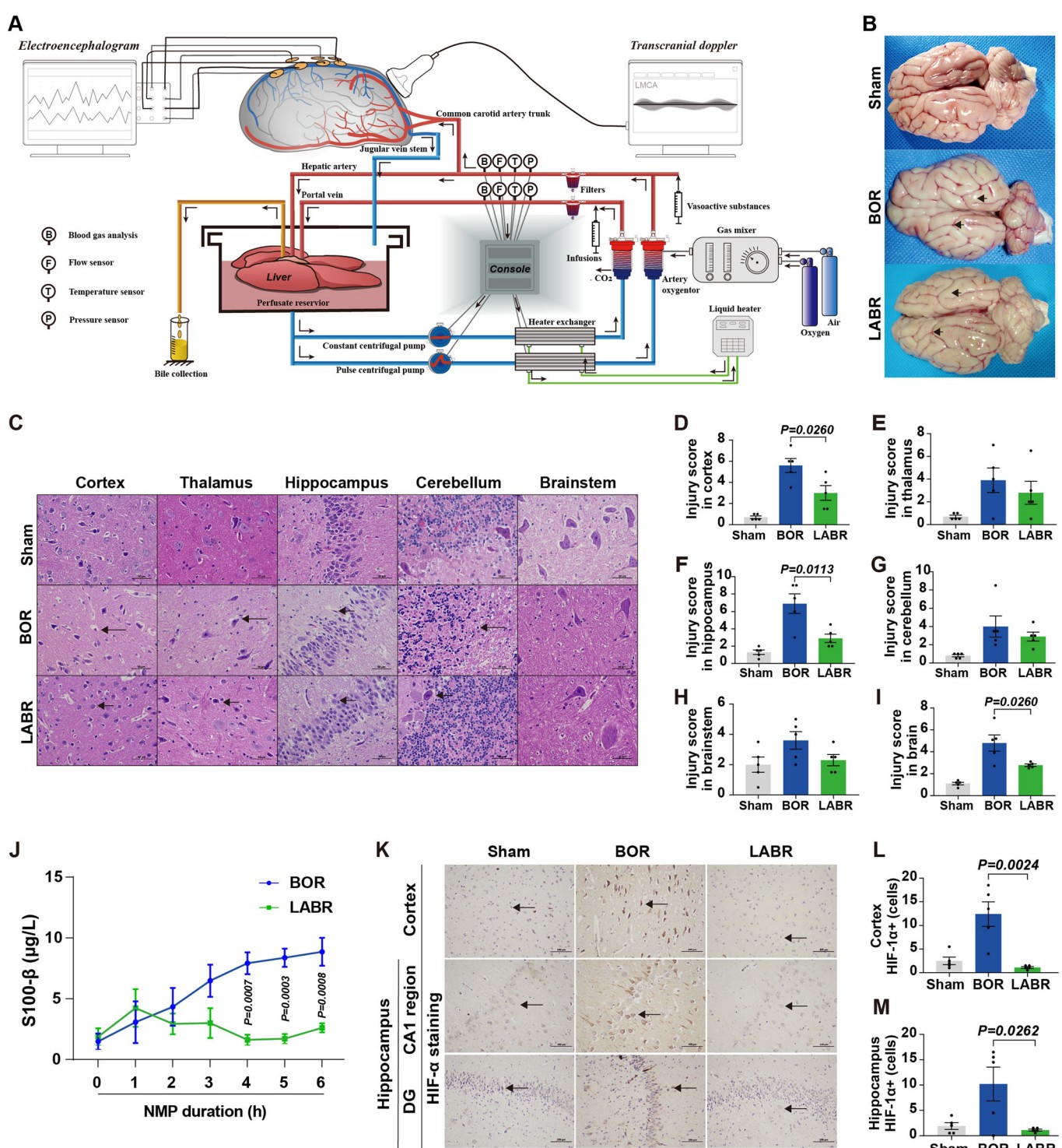

glycerophospholipid pathways were decreased in the BLWI-30 group (Fig. EV4E,F). Both linoleic acid and arachidonic acid metabolism are crucial energy metabolism pathways and are important for brain function. Linoleic acid, arachidonic acid, and their derivatives are essential components in the modulation of synaptic transmission and neurotransmitter release, thereby influencing various neurological processes (Wei et al, 2012). Linoleic acid, arachidonic acid, and their derivatives may

contribute to neuroprotection and support neuronal survival under various pathological conditions, including ischemic injuries (Nayeem et al, 2022). In contrast, the upregulated metabolites in the BLWI-30 group were fewer and enriched in divergent pathways, such as pantothenate and CoA biosynthesis, beta-alanine metabolism, as well as cholinergic synapses (Fig. EV4E,F). The most significantly differentially expressed metabolites are shown in Fig. 6E,F, including variant compounds contributing to

**Figure 3.  Liver-assisted NMP reduces cell damage and hypoxic injury in pig brains.**

(A) Technologies employed in the liver-assisted brain normothermic machine perfusion (NMP) model. (B) Whole brain structure in the three groups. The surface of the brain with cerebral oedema and the narrowed sulci (lower arrowhead) and widened gyri (upper arrowhead), in the BOR group. The characteristics were minimal in the LABR group. BOR, brain-only control with rapid NMP; LABR, liver-assisted brain with rapid NMP. (C) Hematoxylin-eosin staining (400×) of the cerebral cortex, thalamus, hippocampus, cerebellum, and brainstem after 6 h of NMP (arrowheads pointing to the structures that indicate the neuronal shrinkage after suffering ischemia-reperfusion injury) in the five regions. (D–I) The mean injury scores of two fields in the cerebral cortex, thalamus, hippocampus, cerebellum, brainstem or brain (average score of the cerebral cortex, thalamus, hippocampus, cerebellum, and brainstem). (J) The perfusate levels of S100-β. (K) Immunohistochemistry analysis of hypoxia inducible factor-1α (HIF-1α) indicating the degree of hypoxic injury in the hippocampus and cortex. The black arrowhead pointed to HIF-1α positive cells. DG, dentate gyrus. (L, M) Number of cells with upregulated expression of HIF-1α were counted in the cerebral cortex (L) and hippocampus (M). (D–I, J, L, M) $n = 5$, Mean ± SEM; (D–I, L, M) two-tailed ratio unpaired $t$-test; (J) 2-way ANOVA. All replicates shown were biological replicates. Source data are available online for this figure.

energy metabolism and neuronal functions. The distinct metabolome observed in brains experiencing ischemic injury in the presence or absence of concurrent hepatic ischemia highlights the critical influence of the liver on brain metabolic activities under ischemic conditions.

## Metabolomic differences in the perfusate during ex vivo brain NMP with and without support of a functioning liver

The analysis of the metabolomic differences in our global cerebral ischemia model, with or without liver ischemia, revealed significant metabolic differences. These differences were influenced by multiple factors, including variations in energy metabolism differences as well as the immune responses to cell deaths. To investigate the liver's potential contribution to brain energy metabolism, we further conducted ultra-high-performance liquid chromatography coupled to triple-quadrupole mass spectrometry (UHPLC-QqQ-MS) to reveal the perfusate metabolome profiles in the BOR and LABR groups after 2–3 h of NMP. Both the orthogonal projections to latent structures-discriminant analysis (OPLS-DA) (Fig. EV5A,B) and differential analysis showed different metabolome profiles between the two groups (Fig. EV5C,D).

Hierarchical clustering analysis showed the differential metabolites between the BOR and LABR groups (Figs. 7A and EV5E). Interestingly, we found that β-Hydroxybutyric acid (a ketone body) was the most significantly decreased metabolite in the BOR group compared to the LABR group. Moreover, most of the upstream or intermedia metabolites in ketone body metabolism, such as L-Phenylalanine, L-Tryptophan, and Hydroxyisocaproic acid, were significantly increased in the BOR group compared to the LABR group (Fig. 7B–D), suggesting reduced ketone body production in the BOR group. When glucose availability is limited, the liver produces ketone bodies, such as β-Hydroxybutyric acid and acetoacetate, through the process of ketogenesis (Newman and Verdin, 2017). These ketone bodies serve as alternative energy substrates for the brain, particularly during periods of fasting. These results indicates that during ex vivo brain NMP, the liver supports the brain by enhancing energy metabolism through increased production of ketone bodies (Newman and Verdin, 2017).

## Discussion

Post-CA brain injury accounts for two-thirds of total deaths in patients with CA who survived the initial 48–72 h (Sandroni et al,

2021). Clinical studies have shown that both pre-CA cirrhosis and post-CA hypoxic liver injury are associated with significantly poorer survival and neurological outcomes (Bunn et al, 2019; Champigneulle et al, 2016; Reinders et al, 2017; Roedl et al, 2019). In addition, primary liver dysfunction such as cirrhosis, acute liver failure, and acute-on-chronic liver failure, as well as secondary liver dysfunction such as hypoxic liver injury and cholestatic dysfunction, are observed in up to 20% of patients in ICU and are associated with significantly increased mortality (de Garibay et al, 2022; Horvatits et al, 2019). The dual blood supply to the liver in these patients may be compromised in numerous ways. However, the clinical association is not sufficient to provide a causal relationship between the liver dysfunction and severity of post-CA brain injury. In the current study, we present direct evidence for the first time that the liver plays a crucial role in the pathogenesis of post-CA brain injury. This was demonstrated by using both an in vivo global cerebral ischemia model with and without simultaneous hepatic ischemia, and an ex vivo brain NMP model with and without liver support.

The pathophysiology of post-CA brain injury involves both initial ischemic injury and secondary reperfusion injury (Sandroni et al, 2021). The primary manifestation of ischemic injury at the cellular level involves the cessation of aerobic metabolism with subsequent depletion of adenosine triphosphate (ATP). Consequently, there is a significant influx of sodium and water into the cells, resulting in intracellular cytotoxic edema. Reperfusion injury comprises intracellular $Ca^{2+}$ overload and activation of the innate immunity, leading to subsequent tissue inflammation. Ischemic stroke tends to cause frontal lobe atrophy, which is associated with late-life depression and cognitive impairment (Chen et al, 2009; Glodzik-Sobanska et al, 2006). On the other hand, depression is a common symptom observed in patients with liver disease. Mice with hepatic ischemia reperfusion injury exhibited depression-like behaviors, and reduced expression of synaptic proteins in the prefrontal cortex (Yang et al, 2024). In this study, we found in the in vivo isolated global cerebral ischemia model that the infarction ratios in the frontal lobe were larger, in the brains with concurrent hepatic ischemia compared to those without. Taken together, we speculate that there is a crosstalk between the liver and the frontal lobe. Moreover, we observed more severe tissue injury characterized by increased intravascular CD45+ cell adhesion in the brains with concurrent hepatic ischemia, particularly in the temporal lobe. This suggests that there may be a liver-temporal lobe (peripheral-central) immune regulatory pathway. In line with our in vivo data, we found that the lack of a liver in the ex vivo brain NMP circuit compromised the cytoarchitectural integrity and electrocortical activity of the

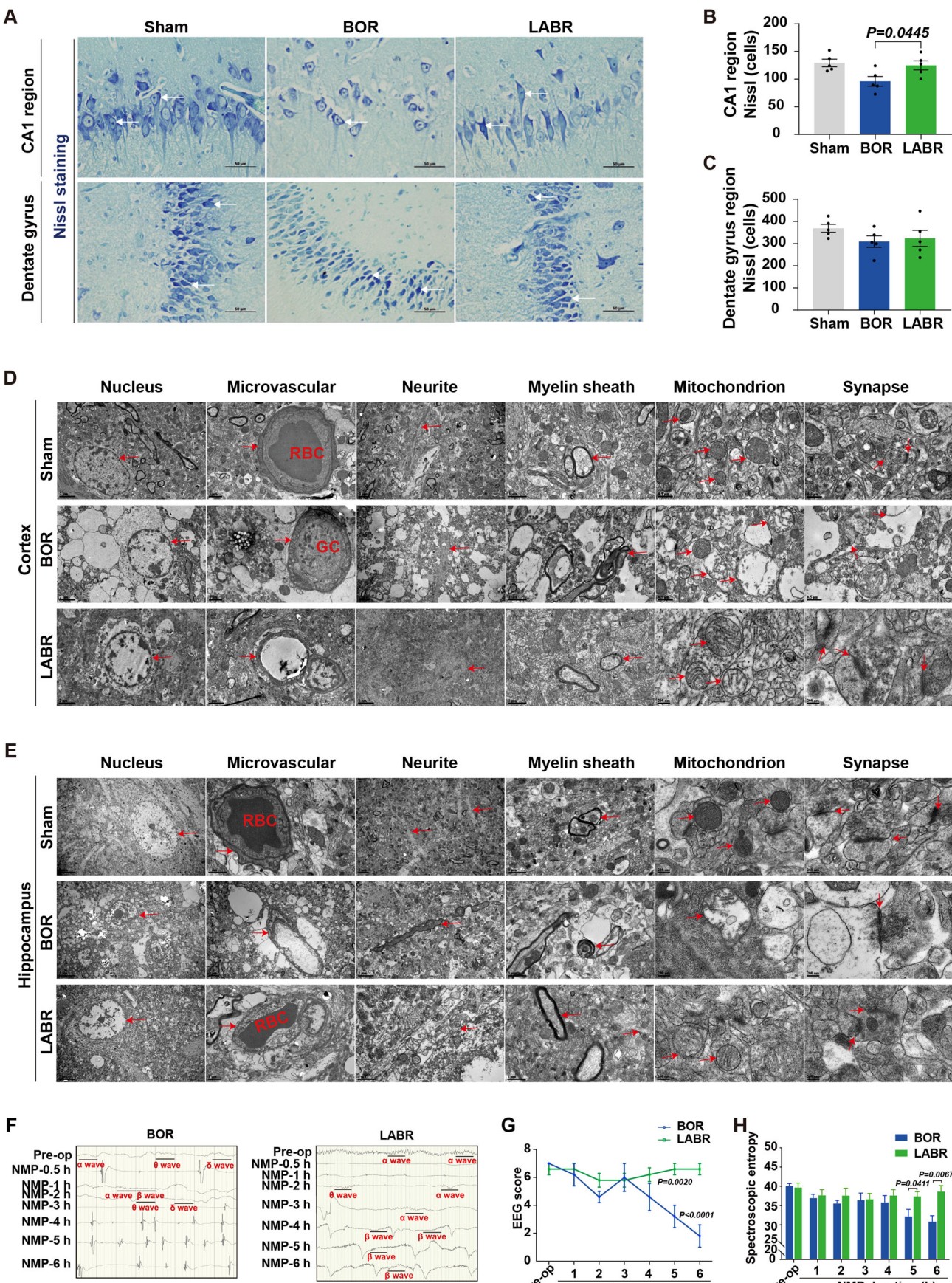

◄ **Figure 4. Improved neuronal viability and electrocortical activity with ex vivo liver-assisted brain NMP compared to brain-only NMP.**

(A) Nissl staining (400×) shows the structural integrity of neuronal somas and axons in the hippocampus of pigs. The white arrowhead pointed to active neurons. BOR, brain-only control with rapid NMP; LABR, liver-assisted brain with rapid NMP. (B, C) Neurons with intact cell bodies were counted in the CA1 region (B) and dentate gyrus (C). (D, E) Transmission electron micrographs of the three groups in the pig cerebral cortex (D) and hippocampus (E). The arrowhead pointed to microvessels, neurites, myelin sheaths, mitochondria or synapses. RBC, red blood cell; GC, granulocyte. (F) Electroencephalography (EEG) results from a representative pig brain in the BOR and LABR groups. (G, H) The EEG scores (G) and spectroscopic entropy (H). (B, G, H) $n = 5$, Mean ± SEM; (B, H) Two-tailed ratio unpaired $t$-test; (G) 2-way ANOVA. All replicates shown were biological replicates. Source data are available online for this figure.

brains in comparison to those with liver support. Collectively, findings from both the in vivo and ex vivo studies show that the liver exerts profound protective effects on post-CA brain injury, particularly in the frontal and temporal lobes.

In addition, our studies on both the transcriptome and metabolome levels provide significant insights into the molecular mechanisms underlying the protective effects of the liver on post-CA brain injury. Our findings indicate that concurrent hepatic ischemia leads to the upregulation of genes associated with cell death and immune responses, and downregulation of genes related to neuronal functions, such as neuron projection development and regulation of neurotransmitter levels, in the brain during ischemic conditions. These molecular alterations are consistent with our functional observations and pathophysiological assays at the cellular level. Moreover, these differentially expressed genes and metabolites might serve as biomarkers for predicting outcomes for patients suffering CA.

Notably, the identification of differentially expressed genes associated with energy reserve metabolic processes underscores the liver's potential role in regulating the metabolism to mitigate the detrimental effects of ischemia on brain tissue. This suggests a possible mechanism through which the metabolic functions of the liver could contribute to the overall protection of the brain during ischemic insults. Previous studies have demonstrated that hepatic soluble epoxide hydrolase (sEH) activity is decreased following traumatic brain injury (TBI) and is negatively correlated with the plasma levels of 14,15-epoxyeicosatrienoic acid, suggesting its neuroprotective effects in the controlled cortical injury mouse model (Dai et al, 2023). Hepatic sEH activity also plays roles in regulating cerebral Aβ metabolism and the pathogenesis of Alzheimer's disease in mice (Wu et al, 2023). Further investigation is needed to identify and validate the specific molecules involved in protecting of the brain during ischemic insults. Such research holds significant clinical relevance for advancing therapeutic interventions.

In clinical practice, various advanced life support technologies have been used to resuscitate patients with CA, including electric defibrillation, mechanical ventilation, extracorporeal membrane oxygenation, and continuous renal replacement therapy (Bateman et al, 2016). The results of the present study showed profound pathological, electrocorticographic, and metabolic impacts of the liver on post-CA brain injury through its regulation of energy metabolism, including the production of ketone bodies. These findings highlight potential therapeutic targets for intervention. To this end, artificial extracorporeal liver support (AELS) might also be a feasible option (Drolz et al, 2011), although the efficacy and duration of AELS are still suboptimal (Larsen, 2019). In addition, extracorporeal support with pig or human liver perfusion

(Chari et al, 1994; Horslen et al, 2000) might be a potential therapeutic option. Recent advances in gene-edited minipigs have made clinical xenotransplantation feasible (Cooper et al, 2020; Griffith et al, 2022; Montgomery et al, 2022; Pierson et al, 2020). These minipigs can provide a timely organ supply for extracorporeal liver support. Moreover, techniques for long-term (7–14 days) ex situ liver NMP have recently been developed (Clavien et al, 2022; Eshmuminov et al, 2020). Therefore, the feasibility and efficacy of extracorporeal support with liver perfusion might be enhanced and translated into clinical practice in the field of post-CA brain injury.

Undoubtedly, there were limitations in this study. Firstly, the study design was exploratory, being observational and not powered to detect specific differences or primary outcomes. The interpretation of the results should be approached with caution. Secondly, spreading depolarization, a clinical marker of early brain injury (Hartings et al, 2017), was not measured in this study. Thirdly, our previous study has shown that the kidney enhances the protective effects of NMP on livers from donation after circulatory death (He et al, 2018). Therefore, the absence of the kidney in both the in vivo and ex vivo models might affect the outcomes of post-CA brain injury. Fourth, only CD45-positive cells were stained in this study, which should be considered as a limitation since individual immune cell types were not measured. In addition, due to the complex function of the liver and the limitations of mechanism studies in pigs, specific genes, proteins, metabolites or immune cells that fully explain the liver's protective effect on the brain could not be identified. However, our data suggests that ketone bodies might be an important target against post-CA brain injury. Subsequently, no therapeutic strategy aimed at enhancing liver function for brain protection was tested in the current study. Finally, the results of the ex vivo electrophysiological activity assessment suggest that the brain might tolerate 50–60 min of WIT, which is significantly longer than the commonly believed of the maximum ischemic tolerance (5–8 min) of the brain. However, the ex vivo assessment of brain function has its limitations. A comprehensive assessment of whole brain function is required, highlighting areas for future studies.

In conclusion, the results from the current study showed the crucial role of the liver in the pathogenesis of post-CA brain injury. These findings shed light on a novel cardio-pulmonary-hepatic-cerebral resuscitation strategy. Moreover, the ex vivo liver-assisted brain NMP model provides a unique platform for further investigating the maximum ischemic tolerance of the brain, and the roles of other organs in post-CA brain injury. Therefore, the insights gained from the current and future studies have the potential to enhance survival and improve outcomes for patients experiencing CA.

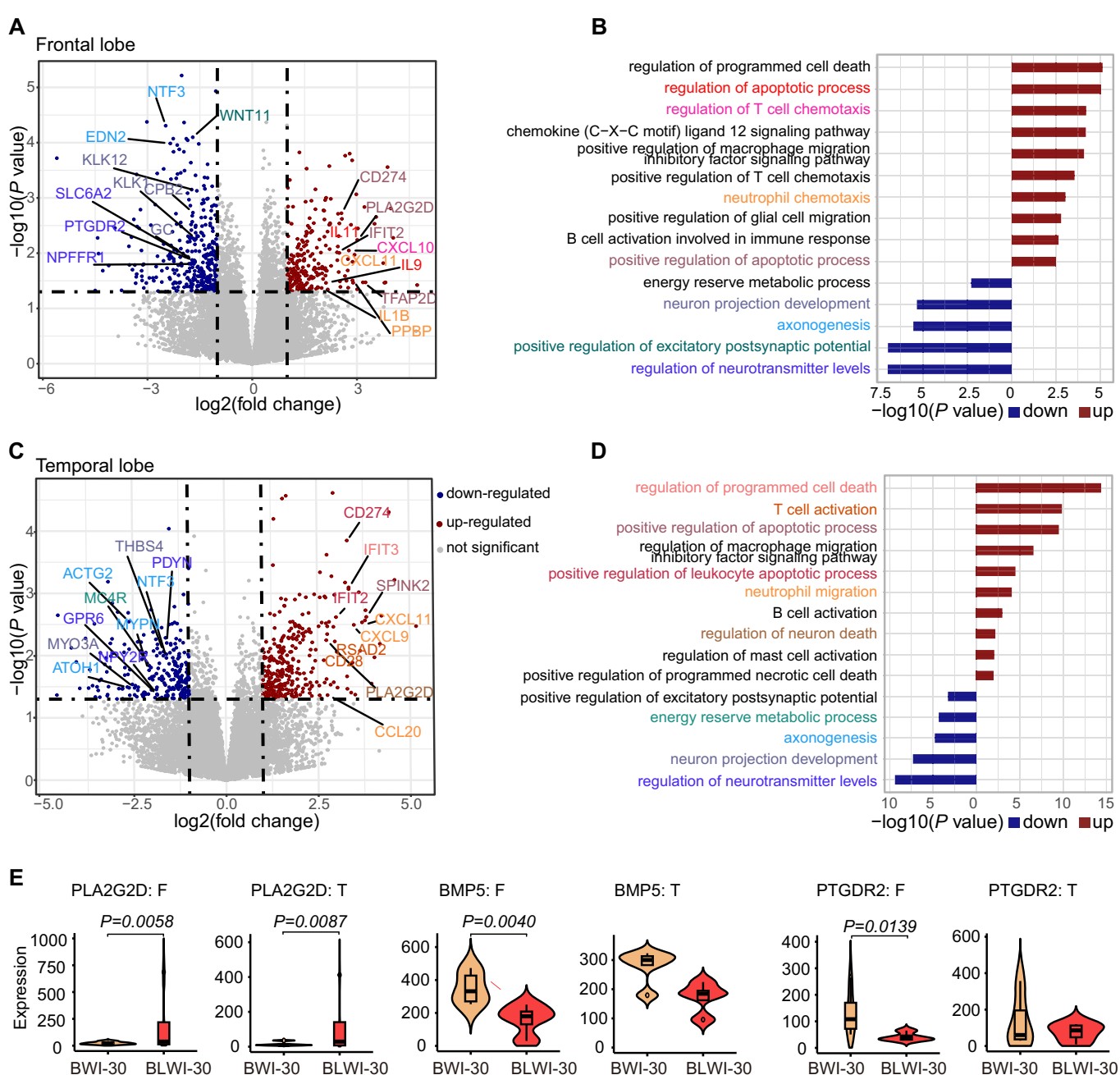

Figure 5. Transcriptomic differences between brain tissues with and without simultaneous hepatic ischemia by RNA-seq.

(A) Volcano plot displaying the differential genes in the frontal lobe of the pig brains with (BLWI-30 group) and without (BWI-30 group) simultaneous hepatic ischemia. Genes significantly upregulated or downregulated in the BLWI-30 group (P value < 0.05, log2(fold change) > 1) are represented in red and blue, respectively. Functional interest genes were annotated and labeled according to their functional terms, as shown in (B). (B) Bar plot illustrating GO enrichment of significant differential genes in the frontal lobe. Upregulated and downregulated genes were tested separately and represented in red and blue, respectively. GO terms were color-coded corresponding to (A). (C) Volcano plot depicting the differential genes in the temporal lobe of the brains of the two groups. Genes significantly upregulated or downregulated in the BLWI-30 group (P value < 0.05, log2(fold change) > 1) are represented in red and blue, respectively. Functional interest genes were annotated and labeled according to their functional terms, as shown in (D). (D) Bar plot displaying GO enrichment of significant differential genes in the temporal lobe. Upregulated and downregulated genes were tested separately and represented in red and blue, respectively. GO terms were color-coded to correspond with (C). (E) Violin plot showing gene expression (RPKM) in each biological replicate. The vertical lines (whiskers) connecting the box represented the maximum and minimum values. The box signified the upper (75th percentiles) and lower quartiles (25th percentiles). The central band inside the box represents the median (50th percentiles). Outliers were shown. F, frontal lobe; T, temporal lobe. (A, C, E) P values and log2(fold change) were calculated by the Wald test using the DESeq2 R package. (B, D) P values were calculated by the hypergeometric test using the MetaCore database. (A–E) Frontal lobe: BWI-30, n = 6, BLWI-30, n = 4; temporal lobe: BWI-30, n = 5, BLWI-30, n = 4. All replicates shown were biological replicates. Source data are available online for this figure.

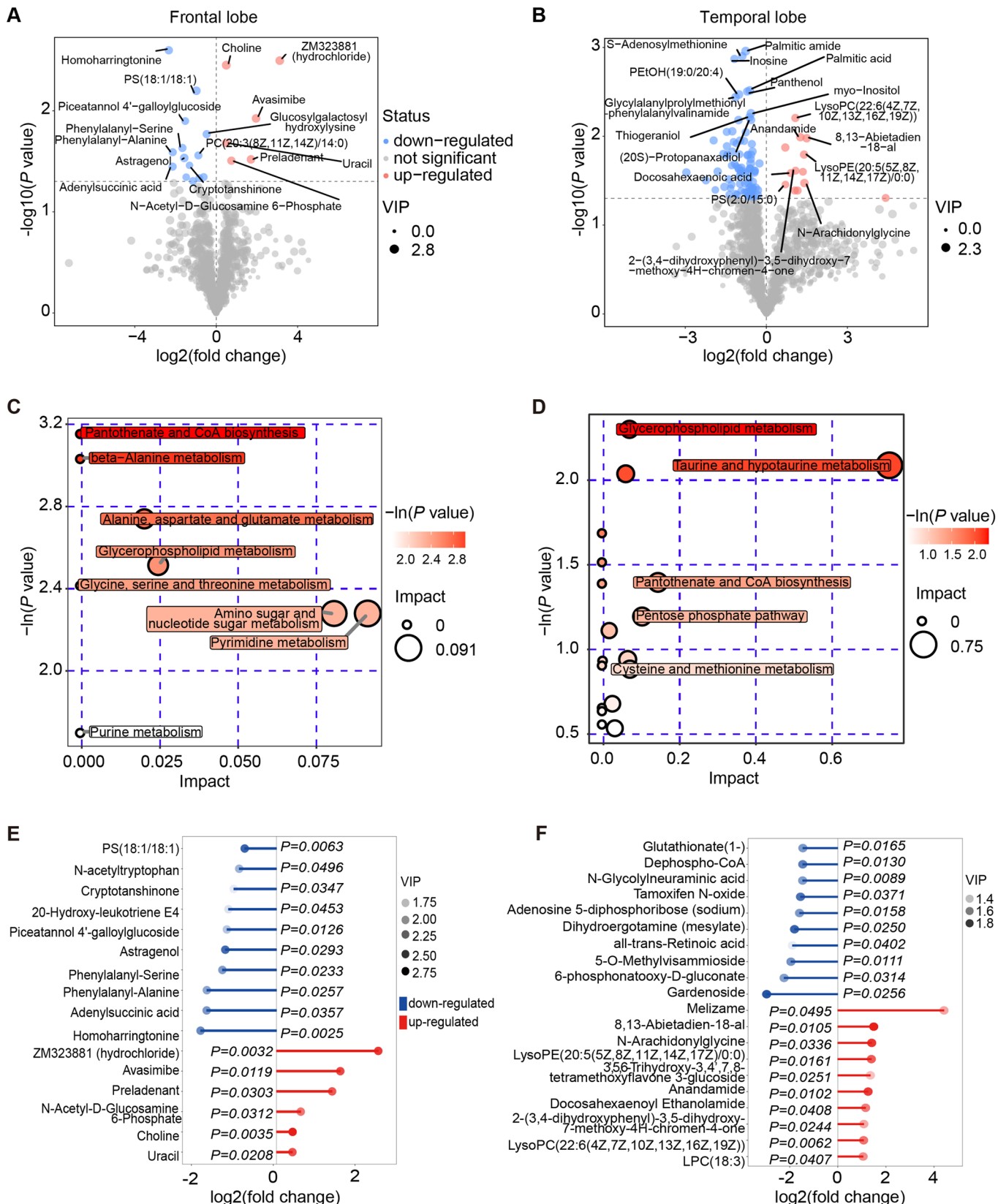

◄ **Figure 6. Metabolic differences in pig brain tissues with and without simultaneous hepatic ischemia by UHPLC-QTOFMS.**

(A, B) Volcano plot illustrating the differential metabolites with secondary mass spectrometry (MS2) names in the frontal lobe (A) and temporal lobe (B) of the pig brains with and without simultaneous hepatic ischemia. Significant metabolic alterations (P value < 0.05, VIP > 1) were denoted in red or blue, respectively. (C, D) Bubble plot displaying the enriched metabolic pathways for increased metabolites in the frontal lobe and temporal lobe, respectively. The value on the x-axis and the size of the circle indicates the impact of the difference. The value on the y-axis and the color indicates the significance of enrichment in $-\ln(P)$. (E, F) Matchstick plot illustrating the top metabolic alterations with lowest P values in the frontal lobe (E) and temporal lobe (F). The x-axis displays the log2(fold change) of BLWI to BWI. Each item's VIP score is denoted by color. (A–F) Frontal lobe: BWI-30, $n = 5$, BLWI-30, $n = 5$; temporal lobe: BWI-30, $n = 6$, BLWI-30, $n = 7$. All replicates shown were biological replicates. P values were calculated by the $t$-test (A, B, E, F) and Fisher's exact test (C, D) using the ggplot2 R package. Source data are available online for this figure.

# Methods

### Reagents and tools table

| Reagent/Resource | Reference or Source | Identifier or Catalog Number |
|---|---|---|
| **Experimental models** | | |
| *Tibet minipigs* | Guangdong Pearl Biotechnology Co. LTD and Southern Medical University | N/A |
| **Recombinant DNA** | | |
| N/A | | |
| **Antibodies** | | |
| Rabbit anti-PTPRC (CD45) | Proteintech | 60287-1-Ig |
| Mouse anti-HIF-1α | Abcam | ab16066 |
| Mouse anti-HSP-70 | Abcam | ab47454 |
| Rabbit anti-NRF-2 | Proteintech | 16396-1-AP |
| Rabbit anti-OGG1 | Abcam | ab233214 |
| Goat anti-IBA1 | Abcam | ab5076 |
| Mouse anti-GFAP | Sigma | G3893 |
| **Oligonucleotides and other sequence-based reagents** | | |
| PCR primers | This study | Appendix Table S1 |
| **Chemicals, Enzymes and other reagents** | | |
| Triphenyl-tetrazolium chloride solution | Asegene | G3005 |
| S100-β Rapid Test Kit | Wuhan Easy Diagnosis Biomedicine | N/A |
| Hematoxylin-eosin solution | Servicebio | G1005 |
| Toluidine blue | Servicebio | G1032 |
| **Software** | | |
| Adobe illustrator 2020 | https://ai.lwb9.cn/?bd_vid=10544275443905869840 | |
| Graphpad Prism v9.4.0 | https://www.graphpad.com/ | |
| **Other** | | |
| MetaCore | https://portal.genego.com/ | N/A |
| BioRender | https://www.biorender.com | N/A |
| KEGG | http://www.genome.jp/kegg | N/A |

## Animals

The feed given to *Tibet minipigs* (female and male) complied with the GB 14924.3-2010 Chinese standard for "Laboratory animals-Nutrients for formula feeds". The pigs were fed in an ordinary environment. Animal facilities had an independent ventilation system, the number of air changes was greater than 8 times per hour, the temperature was controlled between 16 and 26 °C, the daily temperature variation did not exceed 4 °C, and the relative humidity was maintained between 40% and 70%. The living space complied with the GB 14925 Chinese standard for "Laboratory animals—Environment and housing facilities".

## In vivo global cerebral ischemia with or without liver ischemia

Seventeen *Tibet minipigs* (weight, 25.0 ± 3.1 kg; age, 6 months) purchased from Guangdong Pearl Biotechnology Co. LTD were included in this experiment. Seven pigs underwent 30 min of global cerebral and hepatic ischemia followed by reperfusion (BLWI-30 group). Three pigs underwent sham operations (control group). Appendix Fig. S6 summarizes the number of pigs included and excluded in each experimental group for various analyses. Throughout the procedure, the pigs were maintained under general anesthesia via endotracheal intubation, using a combination of inhaled isoflurane (1–2%) and propofol (100 mg/h). Prior to the procedure, pancuronium (0.1 mg/kg) was administered for neuromuscular blockade, and sufentanil (10 μg/kg) was given for perioperative analgesia. Post-operatively, the pigs were monitored for 24 h, unless they died prematurely. Blood and tissue samples were obtained at scheduled time points. Antibiotics were administered every 8 h perioperatively, and body temperature was maintained at 37 °C using a heating pad.

A pig model of global cerebral ischemia and neurological severity scores has previously been established (Allen et al, 2012). Through a 7–8 cm supra-sternal incision, the following arteries were isolated: the innominate artery, left subclavian artery, both internal mammary arteries, both vertebral arteries, both common carotid arteries. The right mammary artery and right external jugular vein were catheterized for the purpose of obtaining blood gas samples and administering drugs. The innominate artery, left subclavian artery just distal to the aortic arch, both internal mammary arteries, and both distal subclavian arteries were clamped for global brain ischemia. Heparin (250 IU/kg) was administered prior to the ischemia insult. All vascular clamps were removed after 30 min of brain ischemia. In vivo Optical Spectroscopy (INVOS) was employed to monitor cerebral tissue oxygen saturation, while ambulatory blood pressure monitoring and neural examination confirmed complete global brain ischemia. Thirty

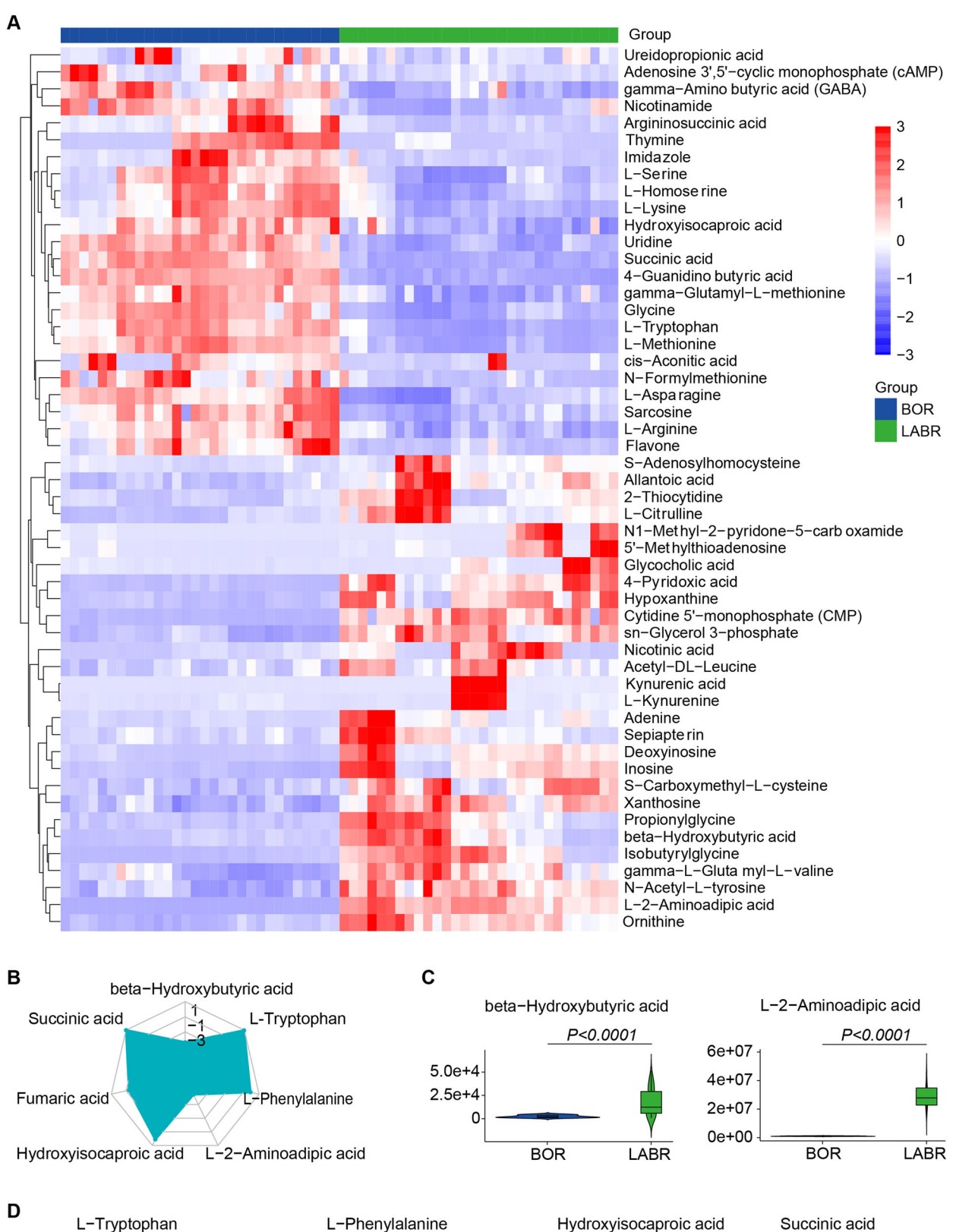

**Figure 7.  Metabolic differences in the perfusate serum during ex vivo pig brain NMP with and without the support of a functioning liver.**

(A) Heatmap displaying significantly differential metabolites (adjusted *P* value (Padj) < 0.05 and |log2(fold change)| > 1), represented by z-score across each row. The heatmap was generated with the pheatmap R package. (B) Radar chart illustrating the differences by metabolic groups. (C) Violin plot showing the signal for decreased metabolites in ex vivo brain normothermic machine perfusion (NMP) without the support of a functioning liver (decreased in the BOR group). (D) Violin plot showing the signal for increased metabolites in ex vivo brain NMP without the support of a functioning liver (increased in the BOR group). (C, D) The vertical lines (whiskers) connecting the box represented the maximum and minimum values. The box signified the upper (75th percentiles) and lower quartiles (25th percentiles). The central band inside the box represents the median (50th percentiles). Outliers were shown. Two-tailed ratio unpaired *t*-test. (A–D) *n* = 5 pigs, triplicate perfusate samples for technical replicates in each pig. Source data are available online for this figure.

minutes after reperfusion, protamine was injected, and the incision was closed. A 20 Fr drainage tube was placed for mediastinal drainage. Midazolam (0.1 mg/kg) was administered in cases where postoperative seizures persisted for more than a few minutes.

In the BLWI-30 group, the portal vein and hepatic artery were thoroughly dissected. The liver underwent simultaneous ischemia and reperfusion with the brain through clamping and unclamping of these two vessels.

## Neurological severity scores

Neurological behavior was independently assessed by two laboratory team members 6 h after reperfusion, using a species-specific behavior scale. The neurological severity scores assess five components as follows (Allen et al, 2012): (I) Central nerve function (0–100 points): pupil size, eye position, light reflex, lid reflex, corneal reflex, ciliospinal reflex, oculocephalic reflex, auditory reflex, gag reflex, and carinal reflex (each 0–10); (II) Respiration (0–100 points): normal (0), hyperventilation (25), abnormal (50), to absent (100); (III) Motor sensory function (0–100 points): stretch reflex, motor response to pain, positioning, and muscle tonus (each 0–25); (IV) Level of consciousness (0–100 points): normal (0), cloudy (30), delirium (45), stupor (60), to coma (100); and (V) Behavior (0–100 points): drinking, chewing, sitting, standing (each 0–15), and walking (0–40). A total score of 0 indicated normal neurological function, while a score of 500 indicates brain death.

## Detection of blood biochemical indicators

Venous blood samples were collected for the detection of biochemical indicators. The blood samples were centrifuged at 3000 rpm for 10 min at 4 °C. Serum samples were collected for aminotransferase (AST) and lactate dehydrogenase (LDH) measurement using the Automatic Chemical Analyzer 7600-100 (Hitachi, Ltd, Tokyo, Japan). Lactic acid and glucose were measured from whole blood samples using a blood gas and chemistry analyzer (EDAN, I15 Vet).

## Triphenyl-tetrazolium chloride (TTC) staining

Frontal coronal brain slices of the pigs were cut with a thickness of 2 mm. The slices were put into 2% triphenyl-tetrazolium chloride solution (Asegene, G3005) in a dark and were bathed at 37 °C for 30 min. After being washed with PBS for 3 min, brain slices were photographed immediately. The white area indicates infarct tissue.

## Surgical procedure for ex vivo brain perfusion

Atropine at a dose of 0.05 mg/kg was administered via intramuscular injection 15 to 30 min before tracheal intubation. Zoletil 50, a combination of tiletamine hydrochloride and zolazepam hydrochloride

(5 mg/kg, Virbac), was administered via intramuscular injection for basic anesthesia. Anesthesia was sustained with inhalation of isoflurane (1–2 vol% end expiratory). Intraoperative infusion of fentanyl (6 µg/kg, repeated every 45 min) was used to maintain analgesia.

After anesthesia, the skin, connective tissues, and musculature were cut around the connection between the C7 and T1 vertebrae with a high-frequency electric knife. The bilateral common carotid arteries and veins were separated, and the vertebral arteries from the subclavian arteries were ligated. Thoracotomy was then performed to expose the roots of the common carotid artery trunk and jugular vein stem. Next, the pigs were intravenously injected with 62500 U heparin. The pigs were euthanized by exsanguination (800–1000 mL of whole blood) via the abdominal aorta, then injected with 1 g of potassium chloride into the left cardiac ventricle to induce cardiac arrest. After cardiac arrest, the common carotid artery trunk and jugular vein stem were separated, and the esophagus, trachea and spinal cord were disconnected.

In the Sham group, immediately after the above surgery, the whole brain was isolated without normothermic machine perfusion (NMP) for pathological assessment. In the brain-only NMP group, a 16 Fr arterial cannula was inserted into the common carotid artery trunk to perfuse the brain. For the liver-assisted brain NMP group, the abdominal aorta (with the celiac artery) was connected to the common carotid artery trunk, and the 16 Fr arterial cannula was inserted into the abdominal aorta to simultaneously perfuse the liver and brain. A straight 24 Fr cannula was inserted into the portal vein to perfuse the liver.

## Ex vivo brain NMP technology

Twenty *Tibet minipigs* (weight, 30–50 kg) purchased from Southern Medical University, Guangzhou, China, were used in the development of the brain NMP technology. In these preliminary experiments, the technology, including the NMP conditions, perfusate components, methods for brain function assessment, and the appropriate ex vivo NMP duration, was optimized. After that, brains from 27 additional pigs were perfused in the formal experiments, and five more brains were harvested immediately after CA without NMP and used as controls (Sham group) in the pathological assessments. In our ex vivo brain NMP system (Fig. 3A), a perfusate based on whole blood (Appendix Table S2) circulated through a pulsatile arterial line and in a nonpulsatile portal vein line. It passed through an oxygenation unit, heater, and filtration unit, after which it perfused the brain and liver. The filtration unit filtered out small blood clots and impurities, and the heater maintained the temperature of the perfusate at 37 °C. Over 10 min, the arterial perfusion pressure was gradually increased from 60 mmHg to 90–95 mmHg to achieve an MCA flow close to the preoperative value. The portal vein perfusion pressure was set at 5–12 mmHg to maintain a flow higher than 500 mL·min⁻¹.

The brains were rapidly reperfused ex vivo or after various intervals. WIT was defined as the interval between cardiac arrest and brain reperfusion. When a liver was perfused simultaneously, it was reperfused ex vivo immediately upon harvesting to ensure its viability.

## Global electroencephalogram (EEG) monitoring of the brains

EEG was monitored during ex vivo brain NMP. After sedation but before tracheal intubation, the initial global EEG of the pig brain was recorded. During the total duration of ex vivo NMP, global EEG activity of the isolated pig heads was continuously recorded. Real-time detection and assessment of global EEG activity from the scalp were performed with a qEEG analysis system (NicoletOne software version 5.94, Natus Medical Inc., San Carlos, CA, USA). The spectral entropy of qEEG was used to assess the conscious states of the isolated pig heads, especially the awake state. Global EEG signals were recorded using 12 silver disc electrodes placed at standard locations: Fp1, Fp2, C3, C4, T3, T4, O1, O2, Fz, Cz, A1, and A2. Based on the presence of δ (1 point), θ (2 points), α (3 points) and β waves (4 points), the EEG score was calculated hourly. Each cerebral hemisphere was scored, and the scores on both sides were averaged.

## Measurement of whole blood gas and biochemistry during the ex vivo brain NMP

Hourly venous perfusate samples were collected from the jugular vein during NMP. Perfusates were collected from the common carotid artery trunk every 20–30 min during NMP. One hundred microliters of each sample were immediately analyzed using the i-STAT®1 Blood Analyzer system (Abbott, Flextronics, Singapore) with CG4+ and EC8+ test cartridges. The arteriovenous gradients of glucose, lactate, pH, and partial pressures of carbon dioxide ($pCO_2$) and oxygen ($pO_2$) were calculated.

## Transcranial Doppler (TCD) cerebral blood flow chart

The initial cerebral blood flow of the pigs was measured after sedation but before tracheal intubation. During ex vivo NMP, cerebral blood flow measurements of the perfused brains were taken every hour. These measurements were conducted on the left middle cerebral artery (MCA) using TCD with a 1.6 MHz probe (EMS-9PB, Delica, Shenzhen, China) through ocular windows at a depth of 4–4.5 cm.

## S100-β measurement

Perfusate samples from the arterial perfusion line were collected and shaken. Samples of 80 mL were applied onto the chip of the S100-β Rapid Test Kit (Immunochromatography) (Wuhan Easy Diagnosis Biomedicine, China). The chip was incubated for 15 min and analyzed in an immunoassay analyzer (QMT8000, Wuhan Easy Diagnosis Biomedicine Co., China).

## Tissue processing and histology

### Detection of brain tissue
After death or following 24 h of in vivo reperfusion, brain tissues were subjected to TTC staining, HE staining, Nissl staining,

IHC, quantitative reverse transcription-polymerase chain reaction (RT-qPCR), RNA sequencing analysis, and metabolomics analysis. Brain tissues after 6 h of ex vivo NMP underwent HE staining, transmission electron microscopy (TEM), Nissl staining, IHC, and immunofluorescence. HE staining, IHC, RT-qPCR, and TEM were performed as described in our previous report (Guo et al, 2022).

### Tissue section preparation
After each experiment, the whole brain was obtained from the head and processed as follows. The cerebral cortex and hippocampus were dissected into 7-mm-thick blocks, fixed in 4% paraformaldehyde, embedded in paraffin, and sectioned into 3–5 μm sections. The sections were mounted on Superfrost Plus slides, and serially dewaxed in dimethylbenzene, alcohol, and ddH$_2$O.

### Hematoxylin-eosin (HE) staining
The tissue sections were counterstained with hematoxylin (Servicebio, G1005, China) for 4 min, rinsed with water, differentiated by hydrochloric acid aqueous solution for 10 s, rinsed again, dyed with aqueous ammonia (Sinopharm Chemical Reagent Co., Ltd, 10002118, China), and then rinsed with water. Next, the sections were dehydrated and embedded in eosin solution (Servicebio, G1005, China) for 5 min. The sections were dehydrated and sealed with neutral gum.

Indicators of brain tissue damage included the disappearance of neurites, cell shrinkage, vacuolization or loosening of tissues, and microvascular damage. The injury scores ranged between 0 and 5 points, representing no injury, very mild injury, mild injury, moderate injury, severe injury, and very severe injury, respectively.

### Nissl staining
The sections were stained with 1% toluidine blue (Servicebio, G1032, China) for 5 min. The sections were quickly rinsed in ddH2O and then differentiated in 1% glacial acetic acid. The degree of differentiation was controlled under the microscope, and the sections were oven dried. The slides were cleared in xylene for 5 min, dried slightly, and sealed with neutral gum.

### Immunohistochemistry
The primary antibodies used included rabbit anti-receptor-type tyrosine-protein phosphatase (PTPRC, CD45) (1:4000; Proteintech; 60287-1-Ig), mouse anti-hypoxia-inducible factor 1-alpha (HIF-1α) (1:100; Abcam; ab16066), mouse anti-70 kDa heat shock protein (HSP-70) (1:200; Abcam; ab47454), rabbit anti-nuclear factor erythroid-derived 2-like 2 (NRF-2) (1:200; Proteintech; 16396-1-AP); and rabbit anti-8-oxoguanine DNA glycosylase (OGG1) (1:150; Abcam; ab233214).

### Immunofluorescence
Antigen retrieval was performed using retrieval fluid (pH = 8.0) (Servicebio, G1202, China) and the sections were washed in PBS (pH = 7.4) (3 × 5 min) before being incubated with an autofluorescence quenching agent for 5 min and washed under running water for 10 min. The slides were blocked in 3% bovine serum albumin for 30 min before incubation with the primary antibody overnight at 4 °C. The primary antibodies used were goat anti-IBA1 (1:100; Abcam; ab5076) and mouse anti-GFAP (1:100; Sigma; G3893). The nuclei were stained with DAPI.

### Microscopy and image processing

Nissl-stained sections were viewed under an upright microscope (Nikon, Eclipse CI, Japan) equipped with an imaging system (Nikon DS-U3). IHC was visualized using a CaseViewer system (3D Histech, Pannoramic MIDI, Hungary), while immunofluorescence was observed with a fluorescence microscope (CIC, XSP-C204, China).

### Histological data analysis and quantification

Images (captured using 10× or 20× objectives) of samples dyed with Nissl, or marker proteins, glial fibrillary acidic protein (GFAP), and ionized calcium binding adapter molecule 1 (IBA1), were standardized to an equivalent image area, randomized, and an investigator blinded to experimental groupings examined the fields of interest. For all images, cells were retained if they exceeded the lowest intensity level, did not have vacuoles larger than the nucleus, or did not exhibit mechanical injury of cells. The images were processed in a random order by an investigator blinded to the experimental group assignments. Objects of interest were counted using the tally function in ImageJ 1.52a (NIH).

## RNA-seq sample preparation, library construction, and data analysis

Total RNA was extracted from the cortical tissues of the frontal and temporal lobes of 13 pigs ($n = 6$ in the BWI group and $n = 7$ in the BLWI-30 group). Each pig served as a biological replicate. Cerebral cortex tissue samples with an RNA Integrity Number (RIN) < 5, indicating poor data quality, were excluded from the study. There were 10 remaining biological replicates in the frontal lobes and nine in the temporal lobes. The cDNA libraries were constructed using a strand-specific RNA-seq protocol. The libraries were subjected to high-throughput sequencing using DNBSeq T7 platforms.

RNA-seq reads were mapped to the reference genome (susScr11) using alignment algorithms (STAR version 2.7.3a), and gene expression levels were quantified using established bioinformatics pipelines (featureCounts Version 2.0.1). Differential gene expression analysis was performed to identify genes exhibiting significant changes in expression between the experimental conditions, using a threshold of log2(fold change) >1 and a $P$ value < 0.05. GO Processes analysis was analyzed using MetaCore.

## Sample preparation for Liquid chromatography-tandem mass spectrometry (LC-MS/MS) from brain tissues

A brain tissue sample weighing 25 mg was placed in the Eppendorf tube, to which 500 µL of extract solution (methanol:acetonitrile:water = 2:2:1, with isotopically labeled internal standard mixture) was added. Then, the samples were homogenized and sonicated thrice in an ice-water bath and incubated for 1 h at −40 °C and centrifuged at 12,000 rpm for 15 min at 4 °C. The resulting supernatant was transferred to a fresh glass vial for analysis. The quality control (QC) sample was prepared by mixing an equal aliquot of the supernatants from all the samples. LC-MS/MS analyses were performed using an UHPLC system (Vanquish, Thermo Fisher Scientific) equipped with a UPLC HSS T3 column (2.1 mm × 100 mm, 1.8 µm) coupled to an Orbitrap Exploris

120 mass spectrometer (Orbitrap MS, Thermo). The ESI source parameters included a capillary temperature of 320 °C, spray voltage set at 3.8 kV (positive) or 3.4 kV (negative), and collision energies of 10/30/60 in NCE mode.

## Metabolites extraction for perfusate serum

Triplicate perfusate samples from the arterial perfusion line were collected after 2 and 3 h of NMP in the BOR and LABR groups. 100 µL of the sample was transferred to an EP tube, and 400 µL extract solution (acetonitrile:methanol = 1:1) containing internal standard (L-2-Chlorophenylalanine, 2 µg/mL) was added. After vortexing for 30 s, the samples were sonicated for 5 min in an ice-water bath. Then, the samples were incubated at −40 °C for 1 h and centrifuged at 10,000 rpm for 15 min at 4 °C. 425 µL of supernatant was transferred to a fresh tube and dried in a vacuum concentrator at 37 °C. Then, the dried samples were reconstituted in 200 µL of 50% acetonitrile by sonication on ice for 10 min. The mixture was then centrifuged at 13,000 rpm for 15 min at 4 °C, and 75 µL of supernatant was transferred to a fresh glass vial for LC/MS analysis. The QC sample was prepared by mixing an equal aliquot of the supernatants from all the samples.

## LC-MS/MS analyses to detect metabolites in the perfusate serum

UHPLC separation was carried out using an Agilent 1290 Infinity series UHPLC System (Agilent Technologies) equipped with a UPLC BEH Amide column (2.1 × 100 mm, 1.7 µm, Waters). The mobile phase consisted of 25 mmol/L ammonium acetate and 25 mmol/L ammonia hydroxide in water (pH = 9.75) (solvent A) and acetonitrile (solvent B). The elution gradient was set as follows: 0–1.0 min, 95% B; 1.0–14.0 min, 95%–65% B; 14.0–16.0 min, 65%–40% B; 16.0–18.0 min, 40% B; 18.0–18.1 min, 40%–95% B; 18.1.1–23.0 min, 95% B. The flow rate was set at 0.5 mL/min. An Agilent 6495 triple quadrupole mass spectrometer (Agilent Technologies) was applied for assay development. The capillary voltage settings were +3000 V for the positive ion mode and −2500 V for negative ion mode. The gas (N2) temperature was maintained at 170 °C with a flow rate of 16 L/min. The sheath gas (N2) temperature was set at 350 °C with a flow rate of 12 L/min. Nebulizer pressure was maintained at 40 psi, and the fragmentor voltage was set at 380 V.

## LC-MS/MS analysis was conducted to analyze metabolites in both in vivo brain tissues and ex vivo perfusate

The data were analyzed using SAS 9.0 (SAS Institute Inc.). The resultant three-dimensional data involving the peak number, sample name, and normalized peak area were fed into SIMCA 14.1 (Sartorius Stedim Data Analytics AB; Umea, Sweden) for principal component analysis (PCA) and OPLS-DA. PCA depicted the distribution of the original data. To obtain an enhanced level of group separation and to better understand the variables responsible for classification, supervised OPLS–DA was applied. This permutation test was conducted to validate the model. Based on OPLS–DA, a loading plot was constructed to illustrate the variables' contributions to the differences between the two groups. To refine this analysis, the first principal

<div style="border:1px solid #ccc; padding:1em;">

**The paper explained**

**Problem**

Sudden cardiac arrest (CA) remains a leading cause of mortality, and brain injury is often followed by post-resuscitation death. Clinical studies suggest that liver function could impact post-CA brain injury, however, no direct evidence has been provided so far.

**Results**

In a pig in vivo model, we observed a larger infarct area in the frontal lobe, elevated tissue injury scores in the CA1 region, as well as increased intravascular CD45+ cell adhesion in the reperfused brains of animals with hepatic ischemia, compared to those without simultaneous hepatic ischemia. Ex vivo, addition of a functioning liver to the brain normothermic machine perfusion (NMP) circuit reduced post-CA brain injury, increased neuronal viability, and improved electrocortical activity. Furthermore, gene expressions and metabolites were affected by the presence or absence of hepatic ischemia.

**Impact**

The current study highlights on a novel cardio-pulmonary-hepatic-cerebral resuscitation strategy, which might help reduce patient death after CA.

</div>

component of VIP was obtained. If $P < 0.05$, then the variable was identified as a significantly differential metabolite between the groups. Using the Kyoto Encyclopedia of Genes and Genomes (KEGG, http://www.genome.jp/kegg), we obtained the metabolic pathways associated with each differential metabolite.

## Exclusion criteria

In the results regarding the infarct area ratio, data from one pig in the BWI-30 group was identified as an outlier according to GraphPad's outlier calculator (https://www.graphpad.com/quickcalcs/grubbs1/). Therefore, the infarct area ratio and all histological data from this pig were excluded from the final analysis in this study. Cerebral cortex tissue with an RNA integrity number (RIN) < 5, indicating poor data quality, was excluded. Outlier data based on PCA were excluded from the metabolomic profile of cerebral cortex tissue. No data were excluded in the ex vivo experiments (Appendix Fig. S6).

## Statistical analysis

All data are presented as either the mean ± SEM or as the median with interquartile range, as appropriate. *T* tests and Mann–Whitney test were conducted for statistical analyses using GraphPad Prism v9.4.0. For comparisons between multiple experimental groups, ordinary 2-way analysis of variance (ANOVA) followed by Sidak's multiple comparison test was employed. $P < 0.05$ was considered statistically significant. When differences were not statistically significant, no indication was reported in the figures. The number of repeats performed and the statistical tests used were showed in the figure legends. Before the experiment, the pigs were randomly grouped. Infarct area measurements, injury scores, and cell counting (active neurons, nucleated blood cells, CD45+ cells, HIF-1α+ cells, HSP70+ cells, NRF2+ cells, and OGG+ cells) were performed blindly.

## Graphics

Figures 1A,B, 3A, S3A, and synopsis graphics were created with BioRender.com, Adobe Illustrator, and Keynote.

## Ethics statement

The study protocol was approved by the Animal Ethical Committee ([2019]193) of the First Affiliated Hospital, Sun Yat-sen University, Guangzhou, China.

# Data availability

The RNA-seq data from this publication have been deposited to the Gene Expression Omnibus database and assigned the identifier. The metabolomics data have been deposited in the MetaboLights, MTBLS10819.

The source data of this paper are collected in the following database record: biostudies:S-SCDT-10_1038-S44321-024-00140-z.

# Peer review information

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

## Acknowledgements

This study was supported by grants from the National Natural Science Foundation of China (81970564, 82170663, 82370664, to Dr Guo; 82070670, to Dr He; 82300744, to Dr Xu), Guangdong Basic and Applied Basic Research Foundation (2024B1515040011 to Dr Guo), the Guangdong Provincial Key Laboratory Construction Projection on Organ Donation and Transplant Immunology (2023B1212060020, to Dr He), Guangdong Provincial International Cooperation Base of Science and Technology (Organ Transplantation) (2020A0505020003, to Dr He), and the Science and Technology Program of Guangdong (2020B1111140003, to Dr He). This study was also supported by the China organ transplantation development foundation (to Dr Guo) and the Medical Scientific Research Foundation of Guangdong Province of China (A2020275 to Dr Yin).

## Author contributions

**Zhiyong Guo**: Conceptualization; Supervision; Funding acquisition; Validation; Investigation; Methodology; Writing—original draft; Writing—review and editing. **Meixian Yin**: Resources; Data curation; Formal analysis; Funding acquisition; Validation; Investigation; Visualization; Methodology; Writing—original draft; Project administration; Writing—review and editing. **Chengjun Sun**: Resources; Data curation; Formal analysis; Investigation; Methodology; Writing—original draft; Project administration. **Guixing Xu**: Data curation; Formal analysis; Investigation; Visualization; Methodology; Writing—original draft. **Tielong Wang**: Investigation; Methodology. **Zehua Jia**: Resources; Investigation; Methodology; Writing—original draft. **Zhiheng Zhang**: Data curation; Investigation; Methodology. **Caihui Zhu**: Data curation; Investigation. **Donghua Zheng**: Investigation. **Linhe Wang**: Resources; Data curation; Investigation. **Shanzhou Huang**: Investigation. **Di Liu**: Formal analysis; Visualization. **Yixi Zhang**: Resources; Investigation. **Rongxing Xie**: Resources; Investigation. **Ningxin Gao**: Formal analysis; Writing—original draft. **Liqiang Zhan**: Resources; Investigation; Visualization. **Shujiao He**: Investigation. **Yifan Zhu**: Data curation. **Yuexin Li**: Data curation. **Björn Nashan**: Writing—review and editing. **Schlegel Andrea**: Writing—review and editing. **Jin Xu**: Formal analysis; Visualization; Methodology. **Qiang Zhao**: Conceptualization; Investigation; Methodology. **Xiaoshun He**: Conceptualization; Supervision; Funding acquisition.

Source data underlying figure panels in this paper may have individual authorship assigned. Where available, figure panel/source data authorship is listed in the following database record: biostudies:S-SCDT-10_1038-S44321-024-00140-z.

## Disclosure and competing interests statement

The authors declare no competing interests.

# Expanded View Figures

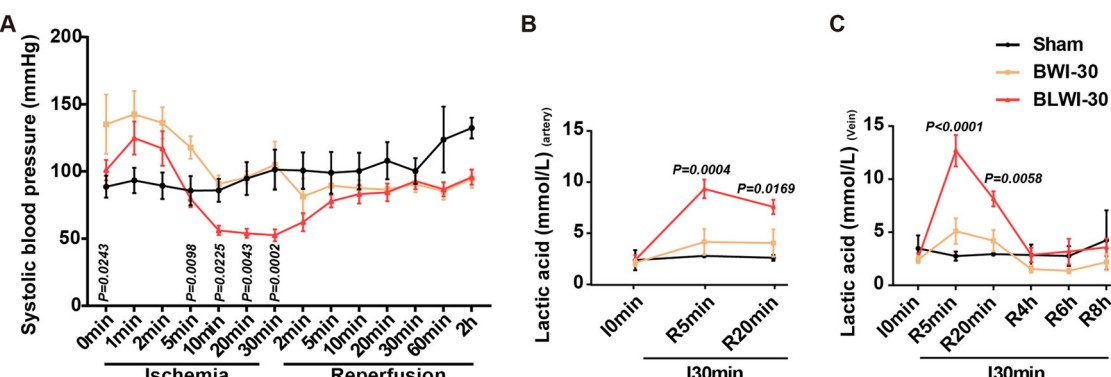

**Figure EV1. Systolic blood pressure and lactic acid levels in the global cerebral ischemia model.**

(A) Systolic blood pressure of the pigs in the BWI-30 and BLWI-30 groups. (B, C) Lactic acid levels of the right mammary artery (B) and right external jugular vein (C) of the pigs in the BWI-30 and BLWI-30 groups. (A–C) Sham, no ischemia, $n = 3$; BWI-30, brain with 30-min warm ischemia, $n = 6$; BLWI-30, brain and liver with 30-min warm ischemia, $n = 7$. All replicates shown were biological replicates. I, ischemia; R, reperfusion. Mean ± SEM, 2-way ANOVA analysis; BLWI-30 versus BWI-30.

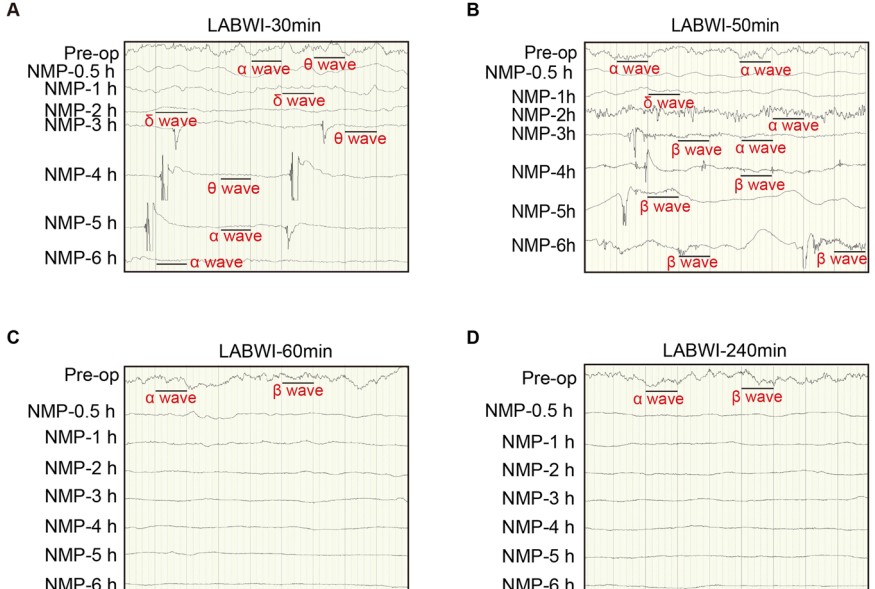

**Figure EV2. Electrophysiological activity during ex vivo liver-assisted brain NMP.**

(A–D) Electroencephalography results of a representative pig brain in the liver-assisted brain groups, where normothermic machine perfusion (NMP) followed intervals of 30-, 50-, 60-, or 240 min. LABWI, liver-assisted brain groups in which brain NMP was preceded by 30–240 min of warm ischemia time; Pre-op, Pre-operation.

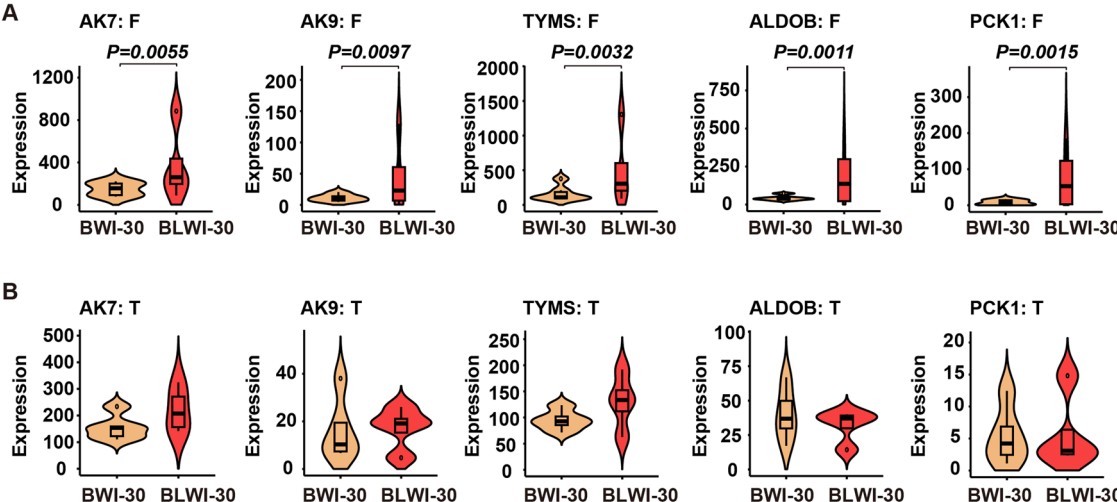

**Figure EV3.  Differentially expressed genes involved in metabolic processes in the global cerebral ischemia model.**

(**A**, **B**) Violin plot illustrating the expression (reads per kilobase per million, RPKM) of differential genes in each of the biological replicates in the frontal lobe (BWI-30, $n = 6$; BLWI-30, $n = 4$) (**A**) or temporal lobe (BWI-30, $n = 5$; BLWI-30, $n = 4$) (**B**) of pigs. All replicates shown were biological replicates. (**A**, **B**) The vertical lines (whiskers) connecting the box represented the maximum and minimum values. The box signified the upper (75th percentiles) and lower quartiles (25th percentiles). The central band inside the box represents the median (50th percentiles). Outliers were shown. *P* values were calculated by the Wald test using the DESeq2 R package.

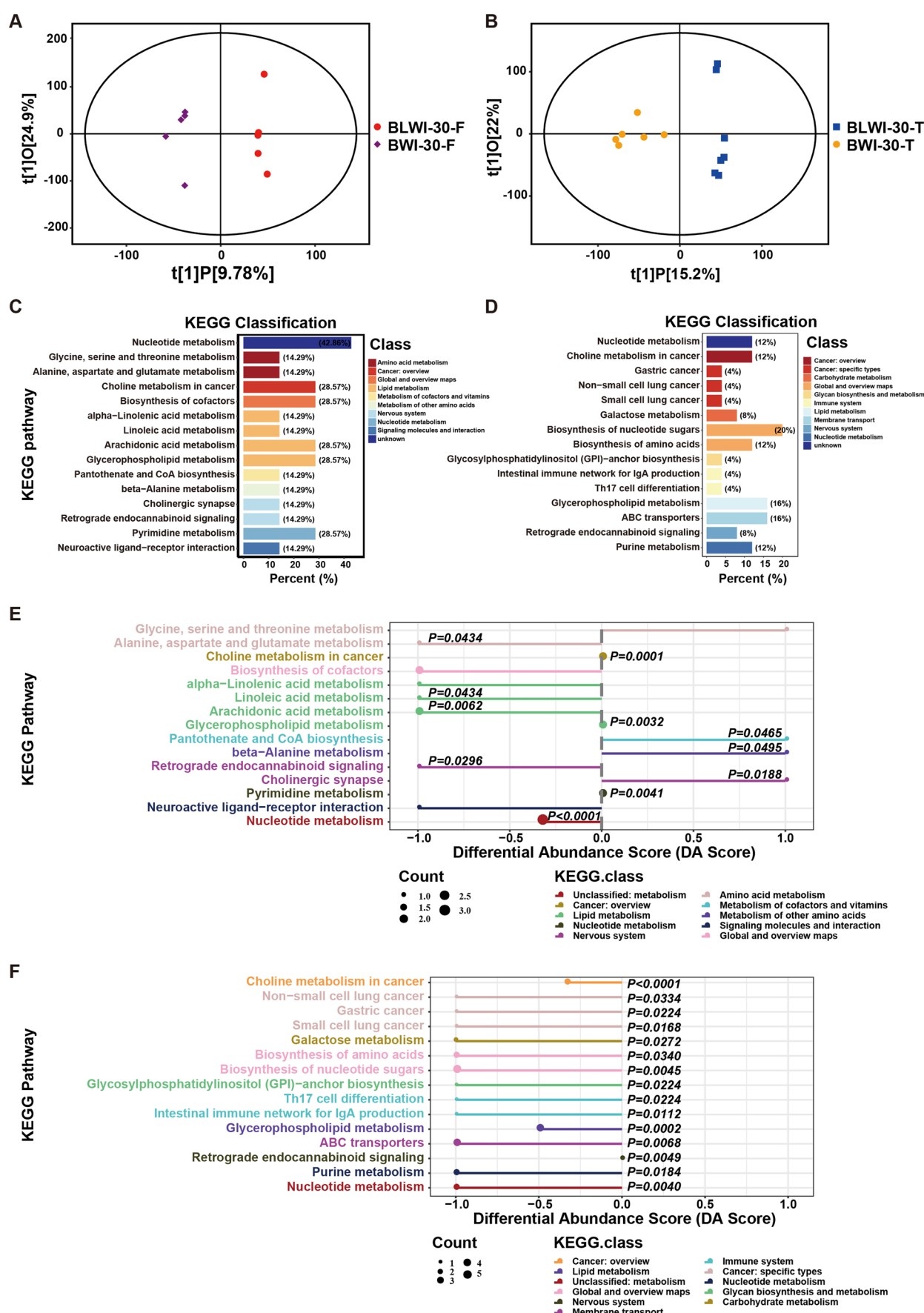

◀ **Figure EV4.  Metabolome differences in pig brain tissues with and without simultaneous hepatic ischemia.**

(A) Scatter plot depicting orthogonal projections to latent structures-discriminant analysis (OPLS-DA) results for the frontal lobe comparing the BLWI-30 and BWI-30 groups. (B) Scatter plot depicting the OPLS-DA data for the temporal lobe comparing the BLWI-30 and BWI-30 groups. (C, D) The bar plot displays the KEGG annotation of detected metabolites for the frontal lobe (C) and temporal lobe (D). The x-axis represents the percentage of identified metabolites in each KEGG class. (E, F) Depiction of overall changes in differential metabolites within a specific pathway for the frontal lobe (E) and temporal lobe (F). The Differential Abundance Score (DA Score) was calculated as the ratio of the difference between the upregulated metabolite count and the downregulated metabolite count on a specific pathway to the total count of metabolites on that pathway. (A–F) Frontal lobe: BWI-30, $n = 5$, BLWI-30, $n = 5$; temporal lobe: BWI-30, $n = 6$, BLWI-30, $n = 7$. All replicates shown were biological replicates. $P$ values were calculated by Fisher's exact test (E, F) using the ggplot2 R package.

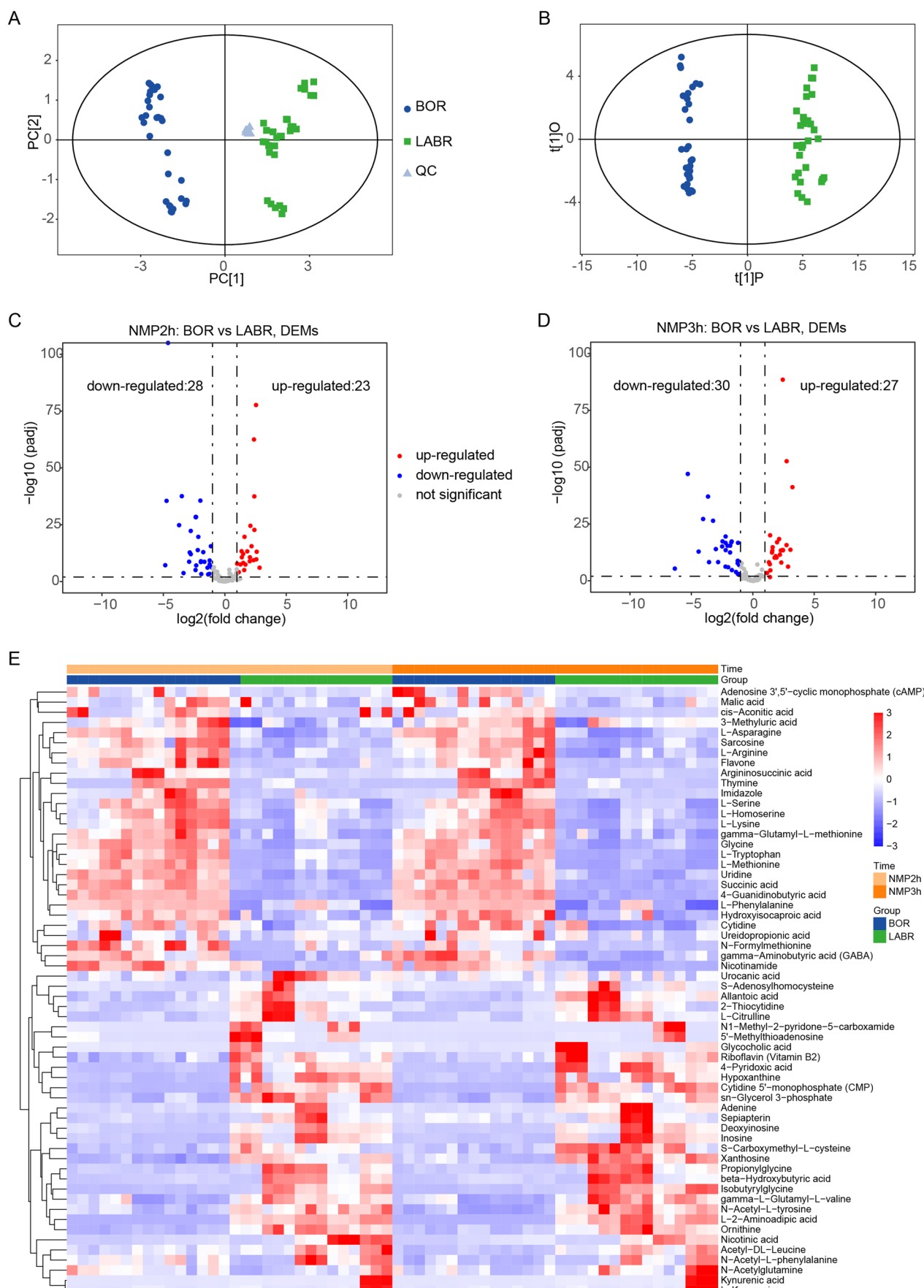

◀ **Figure EV5. Metabolome differences in the perfusate serum during ex vivo pig brain NMP with and without a functioning liver support.**

(A) Principal component analysis (PCA) showcasing the global variation within and between groups for the overall samples. (B) Scatter plot of orthogonal projections to latent structures-discriminant analysis (OPLS-DA) illustrating the differences between groups. (C, D) Volcano plot demonstrating the differential metabolites after 2 and 3 h of normothermic machine perfusion (NMP). (E) Heatmap displaying the z-score distribution of significantly different metabolites across various time points, with each row representing a metabolite. The heatmap was generated with the DESeq2 R package. (A–E) $n = 5$ pigs in each group, triplicate perfusate samples for technical replicates in each pig. (C, D) $P$ values were calculated by the Wald test using the DESeq2 R package.

