## [Peer Review File · EMBO Molecular Medicine]

Liver protects neuron viability and electrocortical activity in post-cardiac arrest brain injury

Zhiyong Guo, Meixian Yin, Chengjun Sun, Guixing Xu, Tielong Wang, Zehua Jia, Zhiheng Zhang, Caihui Zhu, Donghua Zheng, Linhe Wang, Shanzhou Huang, Di Liu, Yixi Zhang, Rongxing Xie, Ningxin Gao, Liqiang Zhan, Shujiao He, Yifan Zhu, Yuexin Li, Björn Nashan, Schlegel Andrea, Jin Xu, Qiang Zhao, and Xiaoshun He

Corresponding authors: Zhiyong Guo (guozhiy2@mail.sysu.edu.cn) , Xiaoshun He (hexsh@mail.sysu.edu.cn), Qiang Zhao (zhaoq37@mail.sysu.edu.cn)

Review Timeline:

Submission Date:	24th Feb 24
Editorial Decision:	29th Apr 24
Revision Received:	7th Jun 24
Editorial Decision:	2nd Jul 24
Revision Received:	18th Jul 24
Accepted:	14th Aug 24

Editor: Lise Roth

Transaction Report:

29th Apr 2024

Dear Prof. Guo,

Thank you for the resubmission of your manuscript to EMBO Molecular Medicine, and please accept my renewed apologies for the delay in getting back to you. We have encountered difficulties securing enough referees for your manuscript, and then obtaining the reports. We have now received the reports from referee #1 (who also reviewed your initial manuscript) and referee #3 (who is a new referee). We have not yet received the report from referee #2 but given that the two other referees provide similar recommendations, and in order not to delay the process further, we decided to make a decision based on the reports at hand. Should referee #2 provide a report shortly, we would send it to you with the understanding that no further reaching experiment would be asked.

As you will see below, both referees are overall supporting publication of your work pending appropriate revisions. Addressing the reviewers' concerns in full will be necessary for further considering the manuscript in our journal. If you would like to discuss further the points raised by the referees, I am available to do so via email or video. Let me know if you are interested in this option.

We are expecting your revised manuscript within three months, if you anticipate any delay, please contact us.

We require:

- 1) A .docx formatted version of the manuscript text (including legends for main figures, EV figures and tables). Please make sure that the changes are highlighted to be clearly visible.
- 2) Individual production quality figure files as .eps, .tif, .jpg (one file per figure). For guidance, download the 'Figure Guide PDF' (<https://www.embopress.org/page/journal/17574684/authorguide#figureformat>).
- 3) At EMBO Press we ask authors to provide source data for the main figures. Our source data coordinator will contact you to discuss which figure panels we would need source data for and will also provide you with helpful tips on how to upload and organize the files.
- 4) A .docx formatted letter INCLUDING the reviewers' reports and your detailed point-by-point responses to their comments. As part of the EMBO Press transparent editorial process, the point-by-point response is part of the Review Process File (RPF), which will be published alongside your paper.
- 5) A complete author checklist, which you can download from our author guidelines (<https://www.embopress.org/page/journal/17574684/authorguide#submissionofrevisions>). Please insert information in the checklist that is also reflected in the manuscript. The completed author checklist will also be part of the RPF.
- 6) It is mandatory to include a 'Data Availability' section after the Materials and Methods. Before submitting your revision, primary datasets produced in this study need to be deposited in an appropriate public database, and the accession numbers and database listed under 'Data Availability'. Please remember to provide a reviewer password if the datasets are not yet public (see <https://www.embopress.org/page/journal/17574684/authorguide#dataavailability>). In case you have no data that requires deposition in a public database, please state so in this section. Note that the Data Availability Section is restricted to new primary data that are part of this study.
- 7) For data quantification: please specify the name of the statistical test used to generate error bars and P values, the number (n) of independent experiments (specify technical or biological replicates) underlying each data point and the test used to calculate p-values in each figure legend. The figure legends should contain a basic description of n, P and the test applied. Graphs must include a description of the bars and the error bars (s.d., s.e.m.). Please provide exact p values.
- 8) Our journal encourages inclusion of *data citations in the reference list* to directly cite datasets that were re-used and obtained from public databases. Data citations in the article text are distinct from normal bibliographical citations and should directly link to the database records from which the data can be accessed. In the main text, data citations are formatted as follows: "Data ref: Smith et al, 2001" or "Data ref: NCBI Sequence Read Archive PRJNA342805, 2017". In the Reference list, data citations must be labeled with "[DATASET]". A data reference must provide the database name, accession number/identifiers and a resolvable link to the landing page from which the data can be accessed at the end of the reference. Further instructions are available at .

9) We replaced Supplementary Information with Expanded View (EV) Figures and Tables that are collapsible/expandable online. A maximum of 5 EV Figures can be typeset. EV Figures should be cited as 'Figure EV1, Figure EV2' etc... in the text and their respective legends should be included in the main text after the legends of regular figures.

10) Author contributions: CRediT has replaced the traditional author contributions section because it offers a systematic machine readable author contributions format that allows for more effective research assessment. Please remove the Authors Contributions from the manuscript and use the free text boxes beneath each contributing author's name in our system to add specific details on the author's contribution. More information is available in our guide to authors.

11) Disclosure statement and competing interests: We updated our journal's competing interests policy in January 2022 and request authors to consider both actual and perceived competing interests. Please review the policy <https://www.embopress.org/competing-interests> and update your competing interests if necessary.

12) Every published paper now includes a 'Synopsis' to further enhance discoverability. Synopses are displayed on the journal webpage and are freely accessible to all readers. They include a short stand first (maximum of 300 characters, including space) as well as 2-5 one-sentences bullet points that summarizes the paper. Please write the bullet points to summarize the key NEW findings. They should be designed to be complementary to the abstract - i.e. not repeat the same text. We encourage inclusion of key acronyms and quantitative information (maximum of 30 words / bullet point). Please use the passive voice. Please attach these in a separate file or send them by email, we will incorporate them accordingly.

13) As part of the EMBO Publications transparent editorial process initiative (see our Editorial at <http://embomolmed.embopress.org/content/2/9/329>), EMBO Molecular Medicine will publish online a Review Process File (RPF) to accompany accepted manuscripts.

In the event of acceptance, this file will be published in conjunction with your paper and will include the anonymous referee reports, your point-by-point response and all pertinent correspondence relating to the manuscript. Let us know whether you agree with the publication of the RPF and as here, if you want to remove or not any figures from it prior to publication. Please note that the Authors checklist will be published at the end of the RPF.

I look forward to receiving your revised manuscript.

Yours sincerely,

Lise Roth

***** Reviewer's comments *****

Referee #1 (Remarks for Author):

An important aspect of enhancing therapeutic approaches following cardiac arrest involves studying the consequences in a gyrated brain model and investigating the influence of the liver as well as the potential effects of ketone body metabolism. The statistical modelling is not ideal but has substantially improved. The framing of the interpretation should consider more the exploratory nature. If a paragraph of limitations clearly states these concerns and if the authors transparently share the number of pigs in total and their usage and mortality, this could be fixed.

Specific points:

"In the in vitro experiments, outlier data of infarct area ratio and related histological data were excluded in each group." This was an exclusion criterion. What does this mean? How did you test for outliers, and what justifies the fact that the data were excluded?

Legend to Figure 1: Sham = 2 does not match the scatter dot plots of the figures.

I do not understand the mortality from a plausible standpoint: "The 24-hour mortality in both BWI-30 and BLWI-30 groups was 28.57% (2/7)". Why is the number of pigs in the BLWI-30 group 7 and in the BWI-30 group 6? Please include a paragraph or supporting figures explaining the rationale for the group design: If 20 pigs were used and 3, 6, and 7 reached the endpoint, adding up to 16 and 2 in each of the ischemia groups (adding to 20), how could the BWI-30 and BLWI-30 groups have each 28,57% mortality? In the case of the BLWI-30 group, it should be 2 out of 9, and in the BWI-30 group, it should be 2 out of 8, given 22.2 and 25% mortality. Even after longer periods of research, it is still unclear how many pigs in total were planned and finally included in which part of the manuscript.

Along these lines, please define biological replicates in the following paragraph: "Total RNA was extracted from the frontal lobes of 10 biological replicates and temporal lobes of 9 biological replicates, followed by the construction of cDNA libraries using a strand-specific RNA-seq protocol. The libraries were subjected to high-throughput sequencing using DNBSeg T7 platforms.". The limitation paragraph in the discussion should clearly state the explorative study design, which was observational and not powered by a specific difference or primary outcome. The overwhelming number of tests without posthoc correction is a source of possible false-positive findings since, by random chance, every 13th t-test will be false-positive. Simply put, figures 1 and 2 include 22 tests without correction. The interpretation of the results should be taken with caution.

Spreading depolarizations can be detected by a subdural electrode strip, even in humans. The authors should at least discuss a lack of SD measurements in their study as a limitation. This is an excerpt from a multi-center trial in Jung Swines: "Here we studied acute sequelae of subarachnoid hemorrhage in the gyrencephalic brain of propofol-anesthetized juvenile swine using subdural electrode strips (electrocorticography) and intraparenchymal neuromonitoring probes." PMID: 28969382

The crosstalk of the liver to the kidneys might affect outcomes and should be discussed, although renal perfusion was not affected in the modeling.

Referee #3 (Remarks for Author):

The severity of brain injury has proven to significantly affect the prognosis of patients suffering from cardiac arrest, and yet the importance of the liver in this process remains largely unexplored. Here, by use of both in vivo and ex vivo pig models to mimic the cerebral ischemia resulting from cardiac arrest, Guo et al. provided compelling evidence showing that a functional liver may assist in reducing infarction area, increasing neuronal viability and ameliorating deficit of electrocortical activity. In addition to their report on the association between the functioning liver and the brain injury, the authors also attempted to reveal the underlying molecular mechanisms by conducting multi-omics assays. Overall, this study is of decent novelty and clinical significance, provides rich omics resources, and hence fits the scope of the journal. Based on their point-to-point responses to Editor and Reviewers, the previous concerns have been adequately addressed. Nevertheless, the manuscript needs further improvements before its being considered for publication.

1. The authors are recommended to re-confirm and articulate the groups and corresponding animal numbers in each experiment. For example, the text (Line 425-427) says that "5 pigs underwent 30 minutes of global cerebral and hepatic ischemia followed by reperfusion (BLWI-30 group), another 5 pigs had global cerebral ischemia and reperfusion (BWI-30 group), and 3 pigs received sham operations (control group)". However, Fig. 1 shows that the n values of the Sham, BWI and BLWI groups equaled to 3, 6 and 7, respectively. Such discrepancies should be explained.

2. More details should be included to make the manuscript more readable and logically sound:

(1) How was the Sham group set up in Fig. 3? Was it a brain-only control, or with support of a functioning liver? Likewise, the grouping criteria for BOR and LABR groups are preferred to be placed in the main text for readers to follow more easily.

(2) Why only the CD45-positive T cells but not other immune cell types were measured?

(3) What could the differences revealed by experiments involving various brain subregions indicate?

(4) Please briefly describe how the neurological severity scores were determined in pig models.

(5) Both the brain and the liver are metabolically active organs. In the ex vivo experiment, how to judge the metabolite changes in the perfusate were attributed to the liver or the brain per se?

(6) Regarding the transcriptomic and metabolic data, could the differentially expressed genes/metabolites be applied as biomarkers predicting the outcome of the patient, or serve as therapeutic targets? This should be further discussed.

3. The language should be further enhanced and proofread. Also, some fonts in the figures are too small to discern, and the symbols (e.g. asterisks indicating the significance) need to be unified throughout the manuscript.

Responses to editors' and reviewers' comments

Editor:

Comment 1

Thank you for the resubmission of your manuscript to EMBO Molecular Medicine, and please accept my renewed apologies for the delay in getting back to you. We have encountered difficulties securing enough referees for your manuscript, and then obtaining the reports. We have now received the reports from referee #1 (who also reviewed your initial manuscript) and referee #3 (who is a new referee). We have not yet received the report from referee #2 but given that the two other referees provide similar recommendations, and in order not to delay the process further, we decided to make a decision based on the reports at hand.

As you will see below, both referees are overall supporting publication of your work pending appropriate revisions. Addressing the reviewers' concerns in full will be necessary for further considering the manuscript in our journal. If you would like to discuss further the points raised by the referees, I am available to do so via email or video. Let me know if you are interested in this option.

Response 1

Thank you for handling our manuscript and supporting publication of our work in your high-impact journal after careful revisions to address the reviewers' concerns. We really appreciate the insightful comments from the reviewers, which help us improve the quality of the manuscript. We have revised the manuscript according to your and the reviewers' suggestion. We hope the revisions can fulfill your high requirement for publication in your journal.

Referee #1 (Remarks for Author):

Comment 1

An important aspect of enhancing therapeutic approaches following cardiac arrest involves studying the consequences in a gyrated brain model and investigating the influence of the liver as well as the potential effects of ketone body metabolism. The statistical modelling is not ideal but has substantially improved. The framing of the interpretation should consider more the exploratory nature. If a paragraph of limitations clearly states these concerns and if the authors transparently share the number of pigs in total and their usage and mortality, this could be fixed.

Response 1

Thank you for your overall positive comments and insightful suggestions. We've revised the manuscript according to your suggestions. We added discussion on the limitations and clearly described the number of pigs in total and their usage and mortality.

Comment 2

2."In the *in vitro* experiments, outlier data of infarct area ratio and related histological data were excluded in each group." This was an exclusion criterion. What does this mean? How did you test for outliers, and what justifies the fact that the data were excluded?

Response 2

Thank you for your comment.

I guess you mentioned the "*in vivo*" instead of the "*in vitro*" experiments.

Outlier laboratory results, specifically those exceeding 3 times the standard deviation from the mean of that specific test if the distribution is normal, can occur. When the outliers are erroneous, they should be removed (Arch Pathol Lab Med 2023;147: 826-836).

In the results of infarct area ratio, the data of one pig in the BWI-30 group was considered as an outlier according to GraphPad's outlier calculator (<https://www.graphpad.com/quickcalcs/grubbs1/>) (see below screenshot). Therefore,

the infarct area ratio (over 97% infarct tissues in the frontal lobe) and all histological data from this pig were excluded for analysis in this study.

We’ve added descriptions on this issue in the revised manuscript. Please view the changes we’ve made in Line179-186, Paragraph 1, Exclusion criteria, Appendix Methods Section, and Appendix Figure 6.

Comment 3

2. Legend to Figure 1: Sham = 2 does not match the scatter dot plots of the figures.

Response 3

Thank you for your comment.

There were 3 pigs in the Sham group. However, one brain sample in the Sham group was not stained by TTC because the purchased reagents did not arrive as planned. Therefore, “Sham, n=2” in panel (D), but “Sham, n=3” in other panels, as stated in the legend to Figure 1: “(D) The infarct area ratio in the frontal lobe of three groups. Sham, n = 2; BWI-30, n = 6; BLWI-30, n = 7;” and “(F-L), (N, O) Sham, n =

3; BWI-30, n = 6; BLWI-30, n = 7". That's why there were 2 dot plots in panel (D) and 3 dot plots in panels (F-L), (N, O) in Figure 1.

Comment 4

3. I do not understand the mortality from a plausible standpoint: "The 24-hour mortality in both BWI-30 and BLWI-30 groups was 28.57% (2/7)". Why is the number of pigs in the BLWI-30 group 7 and in the BWI-30 group 6?

Response 4

Thank you for your comment.

In the *in vivo* experiments, there were 7 pigs in both BWI-30 and BLWI-30 groups. Two pigs in each group died within 24 hours after ischemia-reperfusion. Therefore, the mortality was 28.57% (2/7) in both groups. After surgery, all pig brain tissues were examined. However, as stated in our response to **Comment 2**, the histological data in one pig of the BWI-30 group was excluded for the above mentioned reason in **Response 2**. This explains the discrepancy of the number of pigs in each group for survival analysis and histological analysis.

We've clarified the number of pigs in each group and the exclusion criteria in the revised manuscript.

Please view the changes we've made in Line 429-434, Paragraph 2, Methods Section; Line 179-186, Paragraph 1, Exclusion criteria, Appendix Methods Section; and Appendix Figure 6.

Comment 5

4. Please include a paragraph or Appendix figures explaining the rationale for the group design: If 20 pigs were used and 3, 6, and 7 reached the endpoint, adding up to 16 and 2 in each of the ischemia groups (adding to 20), how could the BWI-30 and BLWI-30 groups have each 28,57% mortality? In the case of the BLWI-30 group, it should be 2 out of 9, and in the BWI-30 group, it should be 2 out of 8, given 22.2 and

25% mortality. Even after longer periods of research, it is still unclear how many pigs in total were planned and finally included in which part of the manuscript.

Response 5

Thank you for your comment.

In our original manuscript, there were 3, 5 and 5 pigs in the Sham, BWI-30 and BLWI-30 groups, respectively. During manuscript revision, 2 pigs each were added to the BWI-30 and BLWI-30 groups. Therefore, in the *in vivo* experiment, there were a total of 3, 7 and 7 pigs in the Sham group, BWI-30 group and BLWI-30 group, respectively. Two pigs died in both BWI-30 and BLWI-30 groups, and that's why the mortality was 28.57% (2/7) in both groups.

We have clarified the number of pigs in each group and how the histological data were excluded in the revised manuscript.

Please view the changes we've made in Line 429-434, Paragraph 2, Methods Section; Line 179-186, Paragraph 1, Exclusion criteria, Appendix Methods Section; and Appendix Figure 6.

Comment 6

5. Along these lines, please define biological replicates in the following paragraph: "Total RNA was extracted from the frontal lobes of 10 biological replicates and temporal lobes of 9 biological replicates, followed by the construction of cDNA libraries using a strand-specific RNA-seq protocol. The libraries were subjected to high-throughput sequencing using DNBSeg T7 platforms."

Response 6

Thank you for your suggestion.

Each pig was a biological replicate. Total RNA was extracted from the cortical tissues of the frontal and temporal lobes of 13 pigs (n = 6 in the BWI group, and n = 7 in the BLWI-30 group). When RNA integrity number (RIN) < 5 indicating a poor data quality, the concerning cerebral cortex tissue were excluded. The remaining

numbers of biological replicates were 10 in the frontal lobes (BWI-30, n = 6; BLWI-30, n = 4) and 9 in the temporal lobes (BWI-30, n = 5; BLWI-30, n = 4).

We explained the issue in the revised manuscript. Please view the changes we've made in Line 471-475, Paragraph 7, Methods Section and Appendix Figure 6.

Comment 7

6. The limitation paragraph in the discussion should clearly state the explorative study design, which was observational and not powered by a specific difference or primary outcome. The overwhelming number of tests without posthoc correction is a source of possible false-positive findings since, by random chance, every 13th t-test will be false-positive. Simply put, figures 1 and 2 include 22 tests without correction. The interpretation of the results should be taken with caution.

Response 7

Thank you for your suggestion.

We've added the limitation on this issue in the revised manuscript. Please view the changes we've made in Line 398-400, Paragraph 6, Discussion Section.

Comment 8

7. Spreading depolarizations can be detected by a subdural electrode strip, even in humans. The authors should at least discuss a lack of SD measurements in their study as a limitation. This is an excerpt from a multi-center trial in Jung Swines: "Here we studied acute sequelae of subarachnoid hemorrhage in the gyrencephalic brain of propofol-anesthetized juvenile swine using subdural electrode strips (electrocorticography) and intraparenchymal neuromonitoring probes." PMID: 28969382.

Response 8

Thank you for your suggestion and reference.

We've added discussion on this limitation in the revised manuscript. Please view the changes we've made in Line 400-402, Paragraph 6, Discussion Section.

Comment 9

8. The crosstalk of the liver to the kidneys might affect outcomes and should be discussed, although renal perfusion was not affected in the modeling.

Response 9

Thank you for your suggestion.

We've added discussion on this issue in the revised manuscript. Please view the changes we've made in Line 402-405, Paragraph 6, Discussion Section.

Referee #3:

Comment 1

1. The severity of brain injury has proven to significantly affect the prognosis of patients suffering from cardiac arrest, and yet the importance of the liver in this process remains largely unexplored. Here, by use of both *in vivo* and *ex vivo* pig models to mimic the cerebral ischemia resulting from cardiac arrest, Guo et al. provided compelling evidence showing that a functional liver may assist in reducing infarction area, increasing neuronal viability and ameliorating deficit of electrocortical activity. In addition to their report on the association between the functioning liver and the brain injury, the authors also attempted to reveal the underlying molecular mechanisms by conducting multi-omics assays. Overall, this study is of decent novelty and clinical significance, provides rich omics resources, and hence fits the scope of the journal. Based on their point-to-point responses to Editor and Reviewers, the previous concerns have been adequately addressed. Nevertheless, the manuscript needs further improvements before its being considered for publication.

Response 1

Thank you for supporting publication of our work and providing us insightful suggestions to improve the quality of manuscript.

Comment 2

2. The authors are recommended to re-confirm and articulate the groups and corresponding animal numbers in each experiment. For example, the text (Line 425-427) says that “5 pigs underwent 30 minutes of global cerebral and hepatic ischemia followed by reperfusion (BLWI-30 group), another 5 pigs had global cerebral ischemia and reperfusion (BWI-30 group), and 3 pigs received sham operations (control group)”. However, Fig. 1 shows that the n values of the Sham, BWI and BLWI groups equaled to 3, 6 and 7, respectively. Such discrepancies should be explained.

Response 2

Thank you for your suggestion.

In our original manuscript, there were 3, 5 and 5 pigs in the Sham, BWI-30 and BLWI-30 groups, respectively. During manuscript revision, 2 pigs each were added to the BWI-30 and BLWI-30 groups. Therefore, in the *in vivo* experiment, there were a total of 3, 7 and 7 pigs in the Sham group, BWI-30 group and BLWI-30 group, respectively. Two pigs died in both BWI-30 and BLWI-30 groups, and that’s why the mortality was 28.57% (2/7) in both groups. In the results of infarct area ratio, the data of one pig in the BWI-30 group was considered as an outlier according to GraphPad’s outlier calculator (<https://www.graphpad.com/quickcalcs/grubbs1/>). Therefore, the infarct area ratio and all histological data from this pig were excluded for analysis in this study. That’s why Fig. 1 shows that the numbers of pigs in the Sham, BWI-30 and BLWI-30 groups equaled to 3, 6 and 7, respectively.

We forgot to update the number of pigs in Line 429-434 and we’ve revised it in the revised manuscript. And we’ve clarified the number of pigs in each group and how some data were excluded in the revised manuscript.

Please view the changes we've made in Line 429-434, Paragraph 2, Methods Section; Line 179-186, Paragraph 1, Exclusion criteria, Appendix Methods Section; and Appendix Figure 6.

Comment 3

3. How was the Sham group set up in Fig. 3? Was it a brain-only control, or with support of a functioning liver?

Response 3

Thank you for your question.

Brains of the Sham group did not undergo normothermic machine perfusion. We've added more descriptions in the revised manuscript, as stated in Paragraph 9, Methods Section that, "five more brains were harvested immediately after cardiac arrest without NMP and used as controls (Sham group)".

Please view the changes we've made in Line 145, Paragraph 6, Results Section; Line 491, Paragraph 9, Methods Section and Line 101-102, Paragraph 13, Appendix Methods Section.

Comment 4

4. Likewise, the grouping criteria for BOR and LABR groups are preferred to be placed in the main text for readers to follow more easily.

Response 4

Thank you for your suggestion.

We've placed the grouping criteria for BOR and LABR groups in the main text in the revised manuscript.

Please view the changes we've made in Line 141-145, Paragraph 6, Results Section.

Comment 5

5. Why only the CD45-positive T cells but not other immune cell types were measured?

Response 5

Thank you for your comment.

In the Figure 2H, the CD45-positive cells represented nucleated hematopoietic cells. It is difficult to find marker antibodies suitable for immunohistochemical staining of various immune cells in formalin fixed tissues of pig brains. We did not find suitable primary antibodies on the websites of companies such as the Abcam and the Cell Signaling Technology. Nonetheless, not measuring the individual immune cell type should be considered as a limitation in this study. We've added this issue as a limitation in the revised manuscript.

We discussed it in Line 405-406, Paragraph 6, Discussion Section.

Comment 6

6. What could the differences revealed by experiments involving various brain subregions indicate?

Response 6

Thank you for your question.

The sensitivity of different brain subregions to ischemia is different. For instance, the CA1 region of the hippocampus (Stroke 2023;54: 673-685. Geroscience 2023;44: 127-130) and the frontal lobe (Clin Neurol Neurosurg 2009;111(10):852-857) are more sensitive to ischemia than other brain regions. Indeed, compared with the BLWI-30 group, the injury scores in these two subregions were lower, the infarct area ratio and the number of CD45+ cells were smaller in the frontal lobe, live neuron counts of the dentate gyrus of the hippocampus was larger, in the BWI-30 versus BLWI-30 group (Figures 1D, F, J, O and Figures 2B).

Ischemic stroke tends to cause frontal lobe atrophy, which is associated with late-life depression and cognitive impairment (Clin Neurol Neurosurg 2009;111(10):852-857. Psychiatry Res 2006;148(2-3):111-120). On the other hand, depression is a frequent symptom in patients with chronic liver disease. Mice with hepatic ischemia/reperfusion injury exhibited depression-like behaviors, and reduced expression of synaptic proteins in the prefrontal cortex (J Affect Disord 2024;15:345:157-167). Taken together, we speculate that there is a crosstalk between the liver and the frontal lobe. In addition, the liver can affect some functional regions of the brain such as the hippocampus and temporal lobe (Science 2020;369(6500):167-173. J Hepatol 1994;21(5):764-770). These are: consistent with our results (Figures 1D, F, H, J and O; 2B and K; 3F and M; 4B and E; 5A-E and 6A-F).

Please view the changes we've made in Line 344-358, Paragraph 2, Discussion Section.

Comment 7

7. Please briefly describe how the neurological severity scores were determined in pig models.

Response 7

Thank you for your suggestion.

We've added descriptions on how we assessed the neurological severity scores in the revised manuscript.

Please view the changes we've made in Line 11-21, Paragraph 2, Appendix Methods Section.

Comment 8

8. Both the brain and the liver are metabolically active organs. In the *ex vivo*

experiment, how to judge the metabolite changes in the perfusate were attributed to the liver or the brain per se?

Response 8

Thank you for your suggestion.

The ischemia time of the brains in the BOR and LABR group was similar. The biggest difference between the two groups was the addition of liver to the perfusion system in the LABR group in comparison to BOR group. Therefore, we believe that the difference of metabolites in the two groups of perfusate serum is mainly caused by the liver. For sure, we can not rule out the possibility that the metabolite changes in the perfusate were attributed to brain per se.

Comment 9

9. Regarding the transcriptomics and metabolic data, could the differentially expressed genes/metabolites be applied as biomarkers predicting the outcome of the patient, or serve as therapeutic targets? This should be further discussed.

Response 9

Thank you for your suggestion.

We've added discussion for the potential of the differentially expressed genes/metabolites as biomarkers to predict the outcomes of the patients or therapeutic targets.

Please view the changes we've made in Line 367-369, Paragraph 3, Line 385-387, Paragraph 5, Discussion Section.

Comment 10

10. The language should be further enhanced and proofread. Also, some fonts in the figures are too small to discern, and the symbols (e.g. asterisks indicating the significance) need to be unified throughout the manuscript.

Response 10

Thank you for your suggestion.

The language has been re-edited by our co-author Björn Nashan, who is a native English speaker. And we have revised the fonts and symbols according to your suggestion.

Please view the changes we've made all through the manuscript and in Figure 1, 2, 4, 7, EV4 and Appendix Figure S1A.

2nd Jul 2024

Dear Prof. Guo,

Thank you for submitting your revised study. We have now received feedback from referees #1 and #3 who were asked to evaluate your revised manuscript. As you will see below, they are overall satisfied with the revisions, and I will therefore be able to accept your manuscript once the following points will be addressed:

1/ Referees' comments: please address the remaining concerns from referee #3 and carefully proof-read the manuscript for grammar and typos.

2/ Manuscript text:

- Please provide institutional email addresses for all corresponding authors.
- Please note that all corresponding authors are required to supply an ORCID ID for their name upon submission of a revised manuscript. ORCID identifiers are currently missing for Qiang Zhao and Xiaoshun He.
- Please remove the highlighted text, and only keep in track changes mode any new modification.
- We can accommodate a maximum of 5 keywords, please adjust accordingly.
- Please remove the Conflict of interest on the first page.
- Methods:
 - o Please remove "The methods used to implement the research will be made available to any researcher for purposes of reproducing the results." and move the methods listed in the supplemental material to the main manuscript file.
 - o Statistics: please detail which experiments were blinded. Please also include a statement on randomization, sample size and inclusion/exclusion criteria
 - o We encourage the inclusion of a Reagents and Tools Table. A downloadable template (.docx) for the Reagents and Tools Table can be found in our author guidelines:
<https://www.embopress.org/page/journal/17574684/authorguide#structuredmethods>
- Data availability: please remove the current text and only provide in this section links to the primary datasets that have been produced in this study and deposited in an appropriate public database (transcriptomics, metabolomics, etc). In case you have no data that requires deposition in a public database, please state so in this section ("This study includes no data deposited in external repositories."). Please correct the checklist "Data availability section" if needed.
- Please move the references after the 'Paper Explained' section.

3/ Figures and Appendix:

- Please provide exact p values, not a range, in the figures or in their legends (including the Appendix).
- Please make sure that all figures and figure panels are referenced in the text. Currently, callouts for Fig 1I, J, K are missing.
- Appendix: please remove the highlighted text. The nomenclature should be corrected to "Appendix Figure S1" and "Appendix Table S1", etc.
- Please address the following queries from our data editors:
 1. Please define the annotated p values **/* as well as provide the exact p-values for the same in the legend of figure 6e-f; EV 4e-f; as appropriate.
 2. Please note that the exact p values are not provided in the legends of figures 1d, f, i-j, l, o; 2b, g, k, n; 3d, f, i-j, l-m; 4b, g-h; 5e; 7c-d; EV 1a-c; EV 3a.
 3. Please indicate the statistical test used for data analysis in the legends of figures 5a-e; 6a-f; 7a; EV 3a-b; EV 4e-f; EV 5c-f.
 4. Please note that in figure EV 3a; there is a mismatch between the annotated p values in the figure legend and the annotated p values in the figure file that should be corrected.
 5. Please note that the box plots need to be defined in terms of minima, maxima, centre, bounds of box and whiskers, and percentile in the legends of figures 5e; 7c-d; EV 3a-b.
 6. Please note that the black arrows are not defined in the legend of figure 1e; 3k. This needs to be rectified.
 7. Please note that the white arrows are not defined in the legend of figure 4a. This needs to be rectified.

4/ Thank you for providing all the source data. We noted duplicated in the source data for Figure 1L, please carefully check these data (see document attached).

5/ I have edited your "Paper Explained", please let me know if you agree with the following, or amend as you see fit:

Problem

Sudden cardiac arrest (CA) remains a leading cause of mortality, and brain injury is often followed by post-resuscitation death. Clinical studies suggest that liver function could impact post-CA brain injury, however no direct evidence has been provided so far.

Results

In a pig in vivo model, we observed a larger infarct area in the frontal lobe, elevated tissue injury scores in the CA1 region, as

well as increased intravascular CD45+ cell adhesion in the reperfused brains of animals with hepatic ischemia, compared to those without simultaneous hepatic ischemia. Ex vivo, addition of a functioning liver to the brain normothermic machine perfusion (NMP) circuit reduced post-CA brain injury, increased neuronal viability, and improved electrocortical activity. Furthermore, gene expression and metabolite levels were affected by the presence or absence of hepatic ischemia.

Impact

The current study highlights a novel cardio-pulmonary-hepatic-cerebral resuscitation strategy, which might help reduce patient death after CA.

6/ Synopsis:

- Please resize the synopsis picture to a tiff/png/jpeg file 550 px wide x 300-600 px high and make sure that the text remains legible.

- I slightly edited your synopsis text, please let me know if you agree with the following or amend as you see fit:

"The liver function in post-cardiac arrest brain injury is not known and was investigated in a pig model of global cerebral ischemia.

- The frontal lobe infarct area and temporal lobe immune cell adhesion were increased in the ischemic brains in the presence of concurrent hepatic ischemia.

- Ex vivo, presence of a functioning liver improved neuronal viability, cytoarchitecture, and electrocortical activity post-CA brain injury.

- Transcriptomics and metabolomics analyses suggest that the liver protects from post-CA brain injury by increased production of ketone bodies."

7/ As part of the EMBO Publications transparent editorial process initiative (see our Editorial at <http://embomolmed.embopress.org/content/2/9/329>), EMBO Molecular Medicine will publish online a Review Process File (RPF) to accompany accepted manuscripts.

This file will be published in conjunction with your paper and will include the anonymous referee reports, your point-by-point response and all pertinent correspondence relating to the manuscript. Let us know whether you agree with the publication of the RPF and as here, if you want to remove or not any figures from it prior to publication.

I look forward to receiving your revised manuscript.

Yours sincerely,

Lise Roth

***** Reviewer's comments *****

Referee #1 (Comments on Novelty/Model System for Author):

An important aspect of enhancing therapeutic approaches following cardiac arrest involves studying the consequences in a gyrated brain model and investigating the influence of the liver as well as the potential effects of ketone body metabolism. The authors have improved their manuscript by adding limitations to their discussion.

Referee #1 (Remarks for Author):

The authors have substantially improved their study, and I have no further concerns.

Referee #3 (Comments on Novelty/Model System for Author):

Following the two rounds of revisions, the technical quality has been further enhanced. Overall, the findings of this study are of broad interest and the data obtained from the pig model are novel and more physiologically relevant to humans; the omics results may also serve as a rich source for future mechanistic investigation.

Referee #3 (Remarks for Author):

In the revised manuscript, the authors elaborated on how the animals were grouped and the corresponding sample sizes; in particular, they explained the inconsistency of animal numbers in different experiments, and provided a new paragraph articulating "Exclusion criteria" as well as a new "Appendix Fig. S6" illustrating the number of pigs used in each experiment and analysis. Further, they added more details, as requested, in the methodology section such as the process of scoring neurological severity; they also discussed the limitations of this study. Based on these revisions, I agree that the manuscript's reliability and readability have been significantly enhanced, which makes it suitable for publication in EMBO Mol Med. However, there remain some grammatic errors and inaccuracies that should be further eliminated during proofreading.

For instance, Page 6, Lines 142-143: "Fig. 3A, Appendix Fig. S3A and Appendix Table S2 demonstrates..."; Appendix Methods, Animal, Page 1, Line 4: "The pigs wers feeded...".

Page 9, Lines 252-253: "We identified 245 up-regulated genes and 350 down-regulated genes in the brain with simultaneous hepatic ischemia (BLWI-30 group) (Fig. 5A)". Indeed, Fig. 5A only shows the result of the frontal lobe but not the whole brain. Appendix Figure 5B, C: "BAI1+" should be corrected to "IBA1+".

Referee #1 (Remarks for Author):**Comment 2**

The authors have substantially improved their study, and I have no further concerns.

Response 2

Thank you for your comment.

Referee #3 (Comments on Novelty/Model System for Author):**Comment 1**

An important aspect of enhancing therapeutic approaches following cardiac arrest involves studying the consequences in a gyrated brain model and investigating the influence of the liver as well as the potential effects of ketone body metabolism. The authors have improved their manuscript by adding limitations to their discussion.

Response 1

Thank you for your positive comments.

Referee #3 (Remarks for Author):**Comment 2**

In the revised manuscript, the authors elaborated on how the animals were grouped and the corresponding sample sizes; in particular, they explained the inconsistency of animal numbers in different experiments, and provided a new paragraph articulating "Exclusion criteria" as well as a new "Appendix Fig. S6" illustrating the number of pigs used in each experiment and analysis. Further, they added more details, as requested, in the methodology section such as the process of scoring neurological severity; they also discussed the limitations of this study. Based on these revisions, I agree that the manuscript's reliability and readability have been significantly enhanced, which makes it suitable for publication in EMBO Mol Med. However, there remain some grammatic errors and inaccuracies that should be further eliminated during proofreading.

Response 2

Thank you for your comments and suggestions.

We re-edited the language with the help from the Elsevier Language Editing Services. Below is the Certificate of Elsevier Language Editing Services.

Certificate for reviewers removed.

Comment 3

For instance, Page 6, Lines 142-143: "Fig. 3A, Appendix Fig. S3A and Appendix Table S2 demonstrates..."; Appendix Methods, Animal, Page 1, Line 4: "The pigs wers feeded...".

Response 3

Thank you for suggestion.

We've changed "demonstrates" to "demonstrate", and changed "wers feeded" to "were fed".

Please view the changes we've made in Line 170, Paragraph 5, Result Section; Line 489, Paragraph 1, Methods Section.

Comment 4

Page 9, Lines 252-253: "We identified 245 up-regulated genes and 350 down-regulated genes in the brain with simultaneous hepatic ischemia (BLWI-30 group) (Fig. 5A)". Indeed, Fig. 5A only shows the result of the frontal lobe but not the whole brain.

Response 4

Thank you for your suggestion.

We changed "brain" to "frontal lobe" in the Line 287, Paragraph 17, Result Section.

Comment 5

Appendix Figure 5B, C: "BAI1+" should be corrected to "IBA1+".

Response 5

Thank you for your suggestion.

We've changed "BAI1+" to "IBA1+" in the Appendix Figure S5B, C.

14th Aug 2024

Dear Prof. Guo,

Thank you for providing your revised files. I am pleased to inform you that your manuscript is now accepted for publication.

However, and as per our email communication, the article will be sent to production to be published online only once the Data availability section will be updated.

Therefore, please send us the updated manuscript file and checklist as soon as possible.

Yours sincerely,
